

# Direct observation of molecular clusters and nucleation mode particles in the Amazon

**Daniela Wimmer[1], Stephany Buenrostro Mazon[1], Hanna Elina Manninen[1,2], Juha Kangasluoma[1], Alessandro Franchin[1,3,4], Tuomo Nieminen[1,5], John Backman[6], Jian Wang[8], Chongai Kuang[8], Radovan Krejci[7], Joel Brito[9,10], Fernando Goncalves Morais[9], Scot Turnbull Martin[11], Paulo Artaxo[9], Markku Kulmala[1], Veli-Matti Kerminen[1] and Tuukka Petäjä[1]**

[1]Department of Physics, University of Helsinki, Gustaf Hallströmin katu 2a, 00560, Helsinki, Finland
[2]European Organization for Nuclear Research (CERN), 1211 Geneva, Switzerland
[3]NOAA Earth System Research Laboratory (ESRL), Chemical Sciences Division, Boulder, CO, USA
[4] Cooperative Institute for Research in Environmental Sciences, University of Colorado Boulder, Boulder, CO, USA
[5]Department of Applied Physics, University of Eastern Finland, Post Office Box 1627, 70211 Kuopio, Finland
[6]Finnish Meteorological Institute, Atmospheric composition research, Erik Palménin aukio 1, 00560, Helsinki, Finland
[7]Stockholm University, Department of Environmental Science and Analytical Chemistry (ACES), 106 91 Stockholm, Sweden
[8]Environmental and Climate Sciences Department, Brookhaven National Laboratory, Upton, New York, USA
[9]Institute of Physics, University of São Paulo, de Fisica, Universidade de Sao Paulo, Rua do Matao 1371, CEP 05508-090, Sao Paulo, Brazil
[10]Laboratory for Meteorological Physics (LaMP), Université Clermont Auvergne, F-63000 Clermont-Ferrand, France
[11]School of Engineering and Applied Sciences, Harvard University Cambridge, Massachusetts 02138, United States of America

*Correspondence to:* Daniela Wimmer (daniela.wimmer@helsinki.fi)

Keywords: atmospheric ions, particle formation, rainforest, high-frequency rainfall

## Abstract

We investigated atmospheric new particle formation (NPF) in the Amazon rainforest using direct measurement methods. The occurrence of NPF on ground level in the Amazon region has not been observed previously in pristine conditions. Our measurements extended to two



field sites and two tropical seasons (wet and dry). We measured the variability of air ion
concentrations (0.8–20 nm) with an ion spectrometer between 2011 and 2014 at the T0t site
and between February and October 2014 at the GoAmazon 2014/5 T3 site. The main difference
between the two sites is their geographical location. Both sites are influenced by the Manaus
pollution plume yet with different frequencies. T0t is reached by the pollution about 1 day in
7, where the T3 site is about 15% of the time affected by Manaus. The sampling was performed
at ground level at both sites. At T0t the instrumentation was located inside the rainforest,
whereas the T3 site was an open pasture site. T0t site is mostly parallel wind to Manaus,
whereas T3 site is downwind of Manaus. No NPF events were observed inside the rainforest
canopy (site T0t) at ground level during the period Sep 2011- Jan 2014. However, rain-induced
ion and particle bursts (hereafter, "rain events") occurred frequently (306/529 days) at T0t
throughout the year but most frequently between January and April (wet season). Rain events
increased nucleation mode (2-20 nm) particle and ion concentrations on the order of $10^4$ cm$^{-3}$.
We observed 8 NPF events at the pasture site during the wet season. We calculated the growth
rates (GR) and formation rates of neutral particles and ions for the size ranges 2-3 nm, 3-7 nm
and 7-20 nm using the ion spectrometer data. One explanation for the absence of new particle
formation events at the T0t site could be a combination of cleaner airmasses and the rainforest
canopy acting as an 'umbrella', hindering the mixing of the airmasses down to the measurement
height. Neutral particle growth rates in the 3-7 nm regime showed two phenomena. Growth
rates were either about 2 nmh$^{-1}$ or about 14 nmh$^{-1}$. There was no clear difference in the sulfuric
acid concentrations for NPF days vs days without NPF. Back trajectory calculations show
different airmass origin for the NPF days compared to non NPF days.

**1 Introduction**

Globally, atmospheric new particle formation (NPF) and growth has been estimated to account
for a major, if not dominant, fraction of atmospheric cloud condensation nuclei (Merikanto et
al. 2009, Wang and Penner, 2009, Yu and Luo, 2009, Dunne et al., 2016; Kulmala et al., 2016).
The formation of atmospheric nanoparticles is a multi-stage process, in which stable clusters
form from gas phase precursors followed by the activation of these clusters for further growth
(Kulmala et al. 2014). Although atmospheric NPF is occurring frequently in many
environments (e.g. Kulmala et al. 2004, Manninen et al. 2010), the Amazon basin is one of the
locations where the initial steps of the formation of nanoparticles have not been previously
observed from ground based measurements (Martin et al, 2010). In the Amazon, emissions and



oxidation of volatile organic compounds (e.g. Lelieveld et al. 2008), aerosol activation to cloud
droplets, and eventually rain formation, are tightly connected and interlinked with
meteorological processes, such as the boundary layer development and deep convection (Wang
et al., 2016). Aerosol concentrations in the atmosphere are rapidly changing with deforestation
and the associated biomass burning and economic development in the Amazon region (Martin
et al. 2016, Artaxo et al., 2013).
The Manaus metropolis (population 2 million) is the capital of the state of Amazonia, Brazil,
surrounded by the largest rainforest on Earth, as shown in Fig 1 (Martin et al, 2017). The
measurements discussed in this paper took place at two different locations in the Amazon
rainforest: a clearing site 70 km downwind from Manaus (T3; Martin et al., 2016), and a site
inside the rainforest canopy, mostly unaffected by Manaus pollution (T0t; Martin et al., 2010b).
The sites will be described in more detail in section 2.1. Depending on the wind direction, these
sites can represent (i) one of the most natural continental locations on Earth, or (ii) a location
affected by both polluted metropolis and rainforest (Martin et al., 2016). The regular synoptic
changes between the wet and dry seasons offered an additional important scientific contrast to
study aerosol dynamics. During most of the wet season the Amazon basin is one of the cleanest
continental regions on Earth (Andreae, 2007; Martin et al., 2010a, Artaxo et al., 2013, Andreae
et al., 2015), while during the dry season biomass burning and local fire emissions are
ubiquitous throughout the basin. Additionally, our study region experiences frequent high-
intensity precipitation episodes.
The primary goal of this paper was to investigate the occurrence of new particle formation
(NPF) and growth in the Amazon region, and to quantify the role of ions and aerosol particles
in this process. No NPF events were observed during the long-term measurements at the site
largely unaffected by Manaus emissions. A clear correlation between rain intensities and ion
concentrations was found for both measurement sites. At the more polluted pasture site, we
observed 8 NPF events, which occurred during the wet season. The data from comprehensive
measurements shows that the freshly formed particles were growing to sizes of about 60 nm at
which they start to act as cloud condensation nuclei.
**2 Methods**
The measurements discussed here were conducted in 2014 outside the rainforest canopy as a
part of the Green Ocean Amazon (GoAmazon2014/5) Experiment (Martin et al., 2016), which



was going on for the period from 1 January 2014 to 31 December 2015. GoAmazon2014/5 was
designed to study the perturbation in cloud and aerosol dynamics by the Manaus emissions.
Our measurement campaign took place during 28 January – 13 October 2014 near the city of
Manacapuru, Brazil, 70 km downwind of Manaus. We compare the campaign data to long-
term measurements made between September 2011 and January 2014.

**2.1. Measurement sites**
**2.1.1 Inside canopy measurements**
The T0t ecological reserve (Martin et al, 2010b) is a terrestrial ecosystem science measurement
site located 60 km north of the Manaus metropolis in the central region of Brazil (-2.609°S, -
60.2092°W). Manaus is situated at the confluence of the Black River (Rio Negro) with the
Solimões river, which together form the Amazon river. The city is an isolated urban region
with a population of more than 2 million people (IBGE, 2015; Martin et al., 2017) and is
surrounded by 1500 km of forests in all directions. T0t is mostly unaffected by the Manaus
pollution and is surrounded by dense rainforest. It allows the characterization of an almost
completely undisturbed natural environment (Martin et al, 2016). The rainforest canopy is
homogeneous with an average height of 30 m.  A Neutral cluster and Air Ion Spectrometer
(NAIS) was placed inside a hut within the rainforest canopy, with an inlet system 2 m above
the ground level. In addition to the ion spectrometer measurements, the measurement hut hosts
a Vaisala system (WXT520) for acquiring meteorological parameters and a differential
mobility particle sizer (DMPS). The DMPS was sampling from an inlet 60 m above the ground
level, therefore sampling aerosols above the canopy. Both the DMPS and the NAIS were
measuring at the T0t site from 2011-2014. For the GoAmazon2014/5 campaign it was moved
to a measurement site outside the rainforest canopy, which is described in Section 2.1.2. The
nucleation and growth rates reported here were all determined from the direct measurements
provided by the NAIS.
**2.1.2 Outside canopy measurements**
T3 is a site equipped with an Atmospheric Radiation Measurement (ARM) Climate Research
Facility of the United States Department of Energy, 70 km downwind of the city of Manaus (-
3.2133°S, -60.5987°W; Mather et al., 2014) and included an ARM Mobile Aerosol Observing
System (MAOS).  The site is located in a pristine environment where the Manaus pollution



plume regularly intersects. Under the day-to-day variability in the meteorology, both clean and
polluted air masses, mixed to variable degrees, arrived at T3. The site is located in a clearing
of the rainforest where the canopy did not hinder mixing. This site also hosted numerous
instrument systems from other GoAmazon2014/5 participants (Martin et al., 2016). The same
NAIS used at T0t was deployed at T3 from end of January 2014 onwards. Sub-3 nm neutral
particle measurements were done with a Particle Size Magnifier (PSM). The PSM and NAIS
inlets sampled at 2 meters from ground level in an open clearing.
**2.2 Instrumentation**
**2.2.1 Neutral cluster and Air Ion Spectrometer (NAIS)**
A Neutral cluster and Air Ion Spectrometer (NAIS; Manninen et al., 2016) was used to
determine the early stages of atmospheric nucleation and subsequent growth. The NAIS
measures the mobility distributions in the range 3.2–0.0013 $cm^2 V^{-1} s^{-1}$, which corresponds to
a mobility diameter range of 0.8–42 nm. The ion and particle size distributions are measured
in three different stages: ion, particle and offset. The NAIS consists of two parallel cylindrical
DMA's (Differential Mobility Analyzers), one for classifying negative ions and the other for
positive ions. When in ion mode, corona chargers and electrostatic filters are switched off to
allow only naturally charged ions into the DMA. During the neutral particle mode, the particles
are charged and then filtered by an electrostatic filter to neutralize them before entering the
DMA. The inlet flow into the NAIS is 60 liters per minute (LPM), whereas the sample and the
sheath flows of the DMA's are 30 and 60 LPM, respectively. The NAIS time resolution was
set to 5 min, where a measurement cycle of negative ions, positive ions, and total particles is
included.
The instrument and calibration are described in more details in Asmi et al. (2009), Wagner et
al. (2016) and Manninen et al. (2016). The accuracy of the ion concentration of the NAIS was
estimated to be 10-30%, which was mainly due to flow rate uncertainties (Manninen et al.,
2016; Wagner et al., 2016). During the campaign, the deposition of particulate matter inside
the instrument caused decreasing flow rates between the maintenance periods. This may have
further increased the uncertainty in measured particle sizes and number especially at sizes
bigger than 20 nm.




### 2.2.2 Particle Size Magnifier (PSM)

The instrument we used to determine aerosol particle concentrations at sizes below 3 nm was a Particle Size Magnifier (PSM; Airmodus A09; Vanhanen et al., 2011). The PSM is a mixing-type condensation particle counter (CPC), in which the aerosol is turbulently mixed with air saturated with diethylene glycol (DEG). DEG only grows the particles to about 90 nm, so the PSM system consists of a second stage, where the particles are grown to optically detectable sizes. The 50% activation diameter of the instrument can be varied in size range 1–4 nm in mobility diameter (Vanhanen et al., 2011) by changing the mixing ratio of the saturator and sample flows. At GoAmazon2014/5, the PSM was used in scanning mode. In scanning mode, the saturator flow is continuously changing, altering the cut-off diameters from 1-4 nm. One scan takes 4 min and the system was setup to do one upscan, followed by a downscan. Due to the challenging measurement conditions, the size resolved data could not be used for the analysis.

Prior to the deployment during the GoAmazon2014/5 campaign, the PSM was equipped with an inlet system, especially designed to decrease the relative humidity of the sample without disturbing the sample itself and maintaining high flow rates (10 Lpm) until the actual sampling to minimize diffusion losses. Laboratory studies have shown that the RH affects the counting efficiency of the PSM drastically (higher sensitivity at smaller sizes at higher RH). The PSM with the inlet was calibrated, using limonene and its oxidation products (Kangasluoma et al, 2014) as the test aerosol. We expect the aerosol sample in Brazil to be mostly organic species, hence the decision to calibrate with limonene. The resulting lowest cut-off diameter of the PSM was 1.5 nm (±0.3 nm). The estimated error is a combination of calibration uncertainty and the influence of the ambient RH on the cut-off diameter of the PSM. To our knowledge, this is the first time when results from ground-based sub-3 nm aerosol particle measurements are shown for the Amazon rainforest. In total, 38 days of data obtained during the dry season were used.

### 2.2.3 Supporting instrumentation at T0t site

Submicron aerosol number size distributions and total particle number concentrations were monitored with a DMPS system (Aalto et al., 2001) and a CPC. The time resolution of the CPC with a cut-off of about 6 nm was 1 minute. The DMPS measured number size distributions over the mobility diameter range of 6–800 nm (Backman et al., 2012). The complete size



distribution is obtained in a 10-minute time resolution. The DMPS system was designed so that the size segregated aerosols were measured for 8 minutes and for the remaining 2 minutes of the 10 min cycle, total particle number concentrations were measured with the CPC directly using a bypass valve. The line losses for the DMPS were estimated to be about 50% for particles smaller than 4 nm in diameter for a similar setup during the AMAZE-08 Experiment (Martin et al., 2010). For the measurements reported here, a similar setup with a 60 m sampling line was used. The DMPS data reported here is qualitative but not quantitative. Formation and growth rates were analyzed using NAIS data only. Meteorological data from a Vaisala weather station included temperature, relative humidity, wind speed and wind direction, and precipitation intensity.

## 2.3 Measurement periods: wet and dry season

The changes between tropical seasons - the wet and dry seasons - offered an additional comparison between two contrasting environmental conditions (Martin et al., 2016, Artaxo et al., 2013). The particle population is in dynamic balance with the ecosystem and anthropogenic contributions (e.g. biomass burning; which produces them directly and indirectly) and the hydrological cycle (which removes them). In the wet season (December to March), the Manaus plume aside, the Amazon basin is one of the cleanest continental regions on Earth (Andreae, 2007; Martin et al., 2010). In the dry and transition season (April to September), biomass burning emissions are prevalent throughout the basin. The most intense biomass burning and atmospheric perturbations take place at the southern and eastern edges of the forest (Brito et al., 2014), however their transport impact the whole basin. Wet deposition decreases whereas condensation sink increases during the dry season. That leads to an overall increase in aerosol concentration in the accumulation mode of about one order of magnitude even in remote areas (Artaxo et al., 2013). Planetary boundary layer development has also a seasonal behavior: stable nocturnal layer and a strong vertical mixing during daytime. The vertical mixing can be enhanced during the wet season due to convective clouds. The nocturnal layer on the other hand traps the emissions near the surface, which can be more pronounced during dry season, as biomass burning usually starts at midday and continues into evening hours (Martin et al, 2010).

## 2.4 Data analysis



All the available data from the NAIS was cleaned for potential instrumental noise. The cleaning
process was done visually using the particle and ion size distributions as surface plots. The
NAIS can measure both naturally charged ions and neutral particle size distributions. We
present data from both measurements in the following. Based on this initial screening, the
decision was made whether one or more of the electrometers was reliable or not and the non-
reliable data was removed. On 44.7% of the days the cleaning procedure was applied. Mostly
the particle data in the smaller size ranges (up to 3 nm) was unreliable. The procedure follows
the guidelines introduced by Manninen et al., 2010. We observed an unexplained increase in
the concentrations of the cluster ions in the NAIS towards the end of October 2013 to January
2014 at the T0t site. This increased level continued when the NAIS was taken to the T3 site.
By comparing the 2014 concentrations of the NAIS channels to those prior to the increase
(January 2012 and 2013), a correction factor of 1.8 was applied to the first 4 NAIS channels
(0.8-1.25 nm) to account for the drift for the subsequent data.
Rain-induced ion events were selected as a day which included an ion burst coincided with the
onset of precipitation. Median and maximum (95 percentile) ion concentrations were calculated
during the time when the rain intensity was $>0$ mm hr$^{-1}$ ($>0.1$ mm hr$^{-1}$ for the T3 site, as the
rain data from the T3 site showed some rain signal on almost all of the days). In some of the
days, rain occurred sporadically several times per day. In order to take this into account, two
separate rain events were classified as such if they occurred $>1$ hr. apart from the end of the
first and the start of the second. Any fluctuations in the rain intensity for a shorter time period
than 1 hr. was considered to be part of a single rain event.
The new particle formation event analysis from the ion spectrometer data, including the event
classification and formation and growth rate calculations, followed the already well-defined
guidelines (Kulmala et al., 2012). In the data analysis, the first step was to classify all available
days into NPF event and non-event days according to methods introduced earlier by Hirsikko
et al. (2007) and Manninen et al. (2010). The days which do not fulfill the criteria of an event
or non-event day, are categorized as undefined days. However, no days were classified as
undefined days in this study. The classification was done visually using daily contour plots of
particle number size distributions. The second step in the analysis was to calculate various
quantities related to each NPF event, such as the particle growth rate (GR) and formation rate
(J). Both growth and formation rates were calculated for three different size bins (2-3 nm, 3-7
nm and 7- 20 nm in diameter) using both ion and neutral particle data from the NAIS. The



particle growth rate was determined by finding the times at which the maximum concentrations
of ions/particles in each of these size bins occurred. A fit between the points was then applied
to determine the GR. The particle formation rate was determined for lower end of each size bin
(2, 3 and 7 nm) by taking into account the growth rates, condensation sink and coagulation
sink.
**3 Results**
All the times mentioned below are local Manaus time (LT), which is Coordinated Universal
Time (UTC) –4 h.
**3.1 Number concentrations of ions and particles at the two sites**
An overview of the observed number concentrations of ions and particles as well as ambient
conditions at the two measurement sites is presented in Table 1. We divided the measured ions
into three sub-size ranges: cluster ions (0.8-2 nm), intermediate ions (2-4 nm) and large ions
(4-20 nm) and the same for neutral particles. The lower and upper limits of the intermediate
ion size range vary in the scientific literature (see Hirsikko et al., 2011 and references therein).
Here, 2-4 nm was chosen, as this size range seems to work well in differentiating between
atmospheric new particle events and non-events when using ion measurements (Leino et al.,
2016). Additionally, the wet and dry seasonality characteristic for the Amazon (Martin et al.
2010) can be observed in the concentration of the large ions (4-20nm): the biomass burning
during the dry season is expected to increase large ion concentrations, whereas during the wet
season their concentrations are expected to decrease due to wet deposition and reduced source
strengths.
Particle and ion concentrations were, in general, higher at the open pasture T3 site, downwind
of Manaus. The average concentrations of 4 – 20 nm particles were a factor of 3 higher in
comparison to parallelwind of Manaus and inside the canopy (T0t). The environmental
variables were relatively similar between the two sites, the temperature and RH being slightly
lower at the outside canopy site compared with the inside canopy site.

**3.1.1 Inside rainforest canopy site (T0t)**



Figure 2 shows the seasonal variability of ions and particles in the three size ranges (0.8-2nm,
2-4 nm and 4-20 nm) for the 2011-2014 period. The cluster ions had a median concentration
of 723 cm$^{-3}$ and 879 cm$^{-3}$ for negative and positive ions, respectively. These medians are higher
than those found in several other locations (eg. urban Paris, Dos Santos et al. 2015; coastal
Mace Head, Vana et al. 2008 and Finokalia, Kalvitis et al. 2012; Puy de Dome, Rose et al.
2016), but comparable to those reported at a boreal forest site in Hyytiälä, Finland (Hirsikko et
al. 2005). Higher cluster ion concentrations have been reported in an Australian rainforest in
Tumbarumba (Suni et al. 2008) and in a wetland site in Abisko (Svennigsson et al., 2008), both
sites having concentrations of ~2400 (1700) cm$^{-3}$ for negative (positive) ions. The size bin of
2-4 nm had a median concentration of 5–10 cm$^{-3}$ for both negative and positive ions. Large
ions 4–20 nm had a median negative (positive) ion concentration of 85 (153) cm$^{-3}$ when
considering the 149 days (out of 524 days), that had data for this size range. These values are
comparable, for example, to intermediate and large ion concentrations found in coastal Mace
Head (Vanta et al. 2008) outside the periods of rain or active NPF. Cluster ion concentrations
are clearly higher in Oct-Dec for both seasons. In general, the positive cluster ion
concentrations are higher in all the cluster ion and intermediate ion size classes for all the
months. Table 2 summarizes the annual concentrations of ions and total particles for the three
size bins.
Differences between the wet (Dec-Mar) and dry and transition season (Apr-Oct) were also
observed in the diel cycle of the ion and particle concentration. Positive and negative cluster
ion concentrations were, on average, higher during the wet season compared to the dry season.
In both seasons, there were more positive than negative cluster ions (Table 1). The lower
concentrations of negative ions are expected due to the Earth's ground 'electrode' effect, in
which negative ions are pushed away from the Earth surface (Hoppel W. A., 1967).
Additionally, cluster ions (0.8-2 nm) showed slightly higher concentrations in the morning and
evening, compared to other times of the day. Enhanced median cluster ion concentrations in
the early morning have also been reported elsewhere, likely due to higher radon concentration
levels at that time of the day (Hirsikko et al 2011 and references therein). A dip in the median
ion concentration after midday coincides with a higher median concentration of large ions,
which is a sign of a larger sink for cluster ions. Intermediate and large ions (2-4nm and 4-20
nm) had only one daytime peak in their concentration in both seasons, similar to the total
particle concentrations shown in Figure 3. For 2–4 nm ions, this occurred in the late afternoon
and was more pronounced during the wet season compared with the dry season for both



polarities. The peak does not seem to be a result of the wet season's rain-induced ion bursts
(Horrak et al. 1998), as discussed in more detail in Section 3.2. Lastly, 4-20 nm ions peaked at
around midday during the wet season, while their diel pattern was more irregular during the
dry season. The negative 4-20 nm ions had the highest concentrations (>1000 cm$^{-3}$) during the
dry season, most likely due to biomass burning and weaker wet deposition. This feature could
not be observed for positive large ions.
The total concentrations of 2-4 nm and 4-20 nm neutral particles had similar daytime peaks
with otherwise stable night-time concentrations (Fig. 3). The median concentration of 2-4 nm
neutral particles was ~500 cm$^{-3}$, which about a factor 100 higher than the median concentration
of 2-4 nm ions. Similar to ions in the same size range, 4-20 nm particles peaked at around
midday, reaching values of about 1000 to 2000 cm$^{-3}$.
**3.1.2. Outside rainforest canopy site (T3)**
The median ion and particle number concentrations during the wet and dry season at the T3
site, outside the canopy and downwind of Manaus, are given in Table 1. The diel cycles of ion
and neutral particle concentrations at this site appeared to be very similar in both wet and dry
season. The cluster ions showed a clear 24 hr cycle: in the mornings (00:00-07:00) their
concentrations were ~1500 cm$^{-3}$ for negative ions, then decreased to ~1000 cm$^{-3}$ and eventual
increased back to ~1500 cm$^{-3}$ after 18:00. This daytime decrease in the concentration is most
likely due to the dilution of the boundary layer. The intermediate ions (2-4 nm) showed higher
concentrations between 03:00 and 06:00 during the dry season compared with the wet season.

The total particle concentration measured by the MAOS CPC (>10 nm total particle
concentration) did not show any diel seasonal cycle. The median total particle concentrations
were about a factor of two higher during dry season (about 1500 cm$^{-3}$) compared with the wet
season (about 700 cm$^{-3}$).

**3.2 Rain-induced ion formation events at inside canopy measurement site**
While no NPF events were observed inside canopy (site T0t), rain-induced ion burst events
(hereafter, rain-events) were common and observed during 306 out the 524 measurement days.
Since multiple rain episodes could occur in a single day, each rain event was investigated
separately, giving a total of 579 rain-events. Figure 4 shows an example of multiple rain-events




that took place during 24 January 2013 (wet season). It is clear from this figure that the negative
ions in the size ranges of 1-3 nm and 3-7 nm increased during the precipitation. A similar
feature for 2-8 nm negative ions during rain events has also been reported for an Australian
rainforest (Suni et al. 2008). Positive ions increased only in the 3-7 nm size range, and showed
even a decrease in the 1-3 nm size range during the time of the precipitation.

Rain-induced bursts are likely a result of a balloelectric effect, in which splashing water
produces intermediate ions such that the negative ions are smaller than the positive ions
(Horrak et al., 2005, Hirsikko et al., 2007, Tammet et al., 2009). The duration of the 579 rain
events varied from a couple of minutes to 22 hours, with over half the rain events lasting for
two hours or less. The rain events were more common during the wet season (Fig. 5) when also
the median rain intensity was higher. Although less frequent, rain-induced particle bursts were
also observed during the dry season. Wang et al., 2016 reported the production of small aerosol
particles, as a result of new particle formation at cloud outflow regions and further transport
into the boundary via strong convection during precipitation events in the Amazon. In the study
by Wang et al. the <20 nm particle concentrations decrease very rapidly, therefore we suggest
the process that we observe to be a local one, as described above. Also the production of ions
we observed only lasted for the duration of the precipitation, whereas Wang et al., observed a
change in the size spectrum that lasted for hours even after the precipitation event.
Figure 6 shows the relation between the median ion concentration and rain intensity during
each rain event. While no clear correlation between these two quantities was found, some
specific features were apparent. First, at the inside canopy site (T0t), the highest cluster ion and
2-4 ion concentrations occurred almost entirely during rather strong rain intensities. Second, at
the site outside the rainforest canopy (T3) shown in the same Figure for comparison, some log-
linear relation between the ion concentration and rain intensity could be observed for rain
intensities >1 mm h$^{-1}$ for all the three size bins.
Rain events were evident also when looking at the total particle concentrations measured by
the NAIS, as depicted in Figure 7. In this example, the rain intensity seemed to have two peaks,
one at ~09:00 followed by a second one at ~11:00. The ion and particle concentrations followed
these two peaks closely. Additionally, the DMPS data showed an appearance of nucleation
mode particles between 6 and 10 nm following the onset of rain. The DMPS was sampling at
a height of 60 m, which is well above the rainforest canopy. The concentration of these 6-10



nm particles increased to ~20 cm$^{-3}$ during the rain event, while being below 5 cm$^{-3}$ throughout
the day outside this peak. The 10-20 nm particle concentration showed first a decrease followed
by a slight increase up to ~35 cm$^{-3}$, peaking later than the 6-10 nm particles. However, it is
unlikely that these are the same rain-induced burst as seen inside the canopy, as the total particle
concentrations seen by the NAIS were of the order of $10^4$ cm$^{-3}$ in the size bin of 4-20 nm. Wang
et al. (2016) reported that nucleation mode particles produced in cloud outflows will be
transported down with the rain, such that they can be observed at the ground level as an increase
in nucleation and Aitken mode concentrations (Dp <50 nm). The appearance of 6-10 nm
particles with its peak concentration, and subsequent increase in the 6-20 nm particle
concentration, could present a similar scenario of small particles brought down from the free
troposphere. However, any possible above-canopy source would be masked by such high rain-
induced concentrations inside the canopy, and it is likely that the dense rainforest canopy would
filter small particles before reaching the ground.

**3.3 New particle formation events at T3**
Table 3 summarizes the overall statistics collected at the outside canopy site (T3). We observed
no NPF events during the dry season, while on 12% of the days during the wet season we could
observe NPF.  Similar event frequency has been observed in Finnish boreal forest environment
during autumn for example (Kontkanen et al., 2017). An earlier study by Backman et al. (2012)
showed that in metropolitan area of São Paulo (population 20 million), Brazil, NPF events
occurred on 18% of the days.
From the NAIS measurements, a total of 113 days were available for the outside canopy
measurements. For the wet season, the data from 28 January until 31 March were used (64
days) and for the dry season the data from 29 August until 13 October was used (46 days). Due
to technical issues, no NAIS data were available for the period 1 April – 28 August 2014. The
PSM measurements were carried out during the dry season only. In total, 38 days of the PSM
data were used and the results are shown in Figure 8. The PSM was used in the scanning mode
but, due to challenging environmental conditions, only the data measured at the highest
supersaturation (total particle concentration at >1.5 nm) is shown.
The PSM shows a similar diel pattern as the cluster ion concentrations measured by the NAIS
(see Fig. 9). We observed a higher median concentration during the early morning (03:00-
06:00), a dip in this concentrations during the early afternoon (12:00-15:00), and then a higher





median concentration in the evening (18:00-24:00). This could be explained by the Carnegie
curve (Harrison, R. G. and Carslaw, K. S., 2003), which shows the diel variation of the
ionospheric potential.
We selected all eight NPF event days to characterize the behavior of ions and aerosol particles
during the particle formation bursts. A comparison of the diel cycle for particles and ions for
nucleation versus no nucleation event days is shown in Figure 9. The cluster ions showed a
clear diel cycle with higher concentrations in the morning and evening both for NPF and non
NPF days. A clear increase in the concentration of the intermediate ions (2-4 nm) occurred
during the NPF event days, which is due to the growth of the ions out of the cluster ion size
range (0.8-2 nm). The intermediate ion concentration increased at around 09:00 LT, suggesting
an onset of the particle formation after sunrise when the boundary layer rises and mixing starts.
The number concentration of 4-20 nm total particles rose within the time window (09:00-12:00
LT) during the nucleation events, while on non NPF days these particles showed highest
concentrations after the sunrise (06:00) and sunset (18:00). The total particle concentration
measured by the MAOS CPC showed a clear concentration increase on NPF event days starting
from 09:00, which clearly indicates that the particles had grown from the smaller sizes to >10
nm, which is the lowest detection limit of the MAOS CPC. No clear diel pattern from the
MAOS CPC measurements was visible on the non NPF days.
The type of NPF events that we observed are likely of regional nature, requiring relatively
homogenous air masses for at least a few hours (Vana et al., 2004, Manninen et al., 2010). The
most likely explanation that no new particle formation events have been observed at the T0t
site compared to the T3 site are either the lack of sources to form sulfuric acid in the more
remote site. Another explanation might be that the sampling at the less polluted T0t site was
performed within the canopy, so the mixing of the boundary layer is hindered by the presence
of the rainforest canopy. The gaps, or fluctuations, in the distinct shape of NPF could be caused
by some degree of heterogeneity in the measured air masses. All the NPF events occurred
during daytime, starting at around 09:00. Sunrise takes place at 06:00 in the Amazon basin. All
the NPF events occurred during the wet season, which might be due to the lower condensation
sink at this time of the year, as shown in Table 1. The median sulfuric acid concentrations as
measured by a quadrupole HOx CIMS (Martin et al, 2016, supplementary material) resulted in
about $9*10^5$ cm$^{-3}$ both for nucleation event days and no NPF days. Similar values have been




reported for Finnish boreal forest measurement site in autumn, where also about 12% of the
days were classified as nucleation event days (Kontkanen et al., 2017).
Back trajectory calculations using HYSPLIT (http://ready.arl.noaa.gov/hypub-
bin/trajtype.pl?runtype=archive) showed a clear difference in airmass origin arriving to the
measurement site between NPF and no NPF days. Back trajectories were calculated 24h
backwards, arriving at 13:00 UTC (09:00 LT) on the NPF days at 500m a. s. l. The same
calculations were performed for each day before/after an NPF day. If an NPF event occurred
on two consecutive days, the day after both events was used for the no NPF day back trajectory
calculcations. On NPF days, the 50th percentile of airmasses originate from about -1.6°S, -
56.5°E and 738.9 m a. s. l. On non NPF days, the back trajectory calculations show origin at -
2.6°S, -56.6°E and 537.4 m a. s. l. These airmasses all originate from upstream of the Amazon
river, where the NPF day airmass originate from further north, which is dense rainforest.
Nevertheless, all airmasses pass over Manaus before reaching the measurement site.
Fig. 10 shows an example of a NPF event observed at the outside canopy (T3) site for both
negative ion and total particle concentration, as measured by the NAIS. The median diameter
of the smallest particle mode measured by the MAOS SMPS decreased at the start of the NPF
event, followed by its continuous growth up to about 60 nm. The intermediate ion
concentrations showed a clear increase during this NPF event, starting at 09:00. Large particles
and ions (4-20 nm) also showed higher concentrations during the event, but with a time delay
of about 30 minutes, indicating the growth of the small clusters to bigger sizes.
Table 4 shows a comparison of the median particle and ion concentrations (25th – 75th
percentiles in brackets), as well as the condensation sink for the time window 09:00-12:00
between the NPF event and non-event days. Clearly, the condensation sink was lower for the
NPF event days (0.001 s$^{-1}$) compared with the non-event days (0.005 s$^{-1}$). We also compared
environmental variables, including the temperature, relative humidity, wind direction and
precipitation. The biggest differences were the factor of 1.6 higher median concentration of
intermediate (2-4 nm) ions for the event days. The environmental variables indicated that there
was no precipitation on any of the classified NPF days, while the median RH was 3% lower
and the median wind direction was 81.6° during event days compared to 105.6° during non-
event days. The temperature was relatively similar between the NPF event and non-event days.



Table 5 shows the calculated GR, particle formation rates and condensation sinks for each of
the classified NPF event day. Both these quantities were determined for three different size
ranges (2-3, 3-7 and 7- 20 nm), and calculated separately for the ion and particle data. The
results show considerably lower ion formation rates compared with neutral particle formation
rates, consistent with observations made at most other continental sites (Manninen et al., 2010;
Hirsikko et al., 2011). The growth rates of particles and ions were comparable to each other
and typically smaller for the 2-3 nm size range compared with the 3-7 and 7-20 nm size ranges.
An increase in the particle/ion growth rate with an increasing particle size has been reported
earlier in a few other sites (see Häkkinen et al., 2013, and references therein).
We observed two regimes when looking at the neutral 3-7 nm GR. On 3 days, the GR were
about 2 nm h$^{-1}$. Those days showed sulfuric acid concentrations of about $2*10^6$ cm$^{-3}$. According
to theoretical calculations about $10^7$ cm$^{-3}$ of sulfuric acid can account for 1 nm h$^{-1}$ (Nieminen
et al., 2010). It is most likely that other compounds are participating to the growth. The other
NPF event days, showed GR of about 14 nmh$^{-1}$ for the same 3-7 nm size range. On those days,
the sulfuric acid concentration was even lower (about $6*10^5$ cm$^{-3}$). These GR are most likely
driven by organic compounds (Tröstl et al., 2016). Tröstl et al. calculated that about $10^6$-$10^7$ of
highly oxidized organic compounds are required to explain GR of about 10 nmh$^{-1}$.
**4 Summary and conclusions**
We performed direct observations of atmospheric new particle formation (NPF) events in the
Amazon area with state-of-the-art aerosol instrumentation. The measurement campaigns were
carried out at two observation sites (T0t and T3) in the vicinity of Manaus city in Brazil. One
of these sites was located inside the rainforest canopy (T0t), providing long-term (Sep 2011-
Jan 2014) measurement data to complement data from outside canopy site (T3).
No NPF events were observed inside the canopy during the period Sep 2011-Jan 2014.
However, we observed rain-induced ion and particle burst events ("rain-events") inside the
canopy during 306 of the 529 days. Concentrations of 2-4 nm and 4-20 nm ions and total
particles were enhanced by up to 3 orders of magnitude during such rain-events (~$10^4$ cm$^{-3}$).
The rain events occurred throughout the year, but were most frequent during the wet season
(January to April) when also the median rain intensity was the strongest. Multiple rain events
could occur during the same day, totaling 579 rain events in 306 rainy days. The duration of
the rain events ranged from a couple of minutes to 22 hours, but over 50% of the events lasted



for <2 hours. Overall, the median positive cluster ion (0.8-2 nm) concentrations was higher
than that of negative cluster ions, as can be expected from the Earth's electrode effect (Hoppel,
W. A., 1967). However, during the rain-events 0.8-2 nm and 2-4 nm negative ions dominated
over similar-size positive ions. Similar, but weaker, rain-events were found at the site outside
the rainforest canopy (T3).
Outside the rainforest canopy, we observed a clear diel pattern in the cluster ion concentration
during both wet and dry season, with higher concentrations during the morning and evening
compared with other times of the day. The results from the PSM showed a very similar pattern:
the median diel cycle of >1.5 nm particles showed a higher concentration in the early morning
and a dip in the afternoon followed by an increase in the evening after sunset. The diel pattern
was less pronounced inside the canopy, which indicates that the rainforest canopy acts as a sink
for newly formed particles and hinders vertical mixing.
We observed eight NPF events showing particle growth at site T3 outside the canopy during
Jan-Oct 2014, which is the wet season. The formation rates were considerably higher for
neutral particles compared with ions during the NPF events. The growth rates of newly formed
ions and particles were comparable to each other and showed a clear increase with increasing
size in the sub-20 nm size range. We found two different regimes for the 3-7 nm size range.
We found 3 out of 8 NPF days with GR of about 2 nm h$^{-1}$ and 4 out of 8 NPF days with GR of
about 14 nm h$^{-1}$. The sulfuric acid concentrations were the same for the nucleation event days
and non-event days (approx. $9*10^5$ cm$^{-3}$). The back trajectory calculations using HYSPLIT did
not show any clear difference between days with small GR compared to days with high GR.
Nevertheless, a clear difference in airmass origin on NPF days compared to the same number
of days without NPF was observed. Most likely the observed growth on all the NPF days is
driven by highly oxidized organic compounds (Tröstl et al., 2016). The back trajectory
calculations show airmass origin over rainforest area on NPF days, vs the Amzon river on days
where no NPF was observed. As shown by the SMPS, the particles grew to sizes of around 60
nm during all the NPF events, which means they are able to act as cloud condensation nuclei
(McFiggans et al, 2006; Andreae and Rosenfeld, 2008; Kerminen et al. 2012). There were clear
differences in median cluster and intermediate ion concentrations between the NPF event days
and non-event days for the time window of 09:00-12:00 LT. The median cluster ion
concentration was lower, and the median intermediate ion concentration was higher by a factor
of 1.6, during the NPF event days compared with non-event days. The condensation sink was





lower during the NPF event days (0.0016) compared with non-event days (0.005). No
precipitation was observed on any of the NPF event days. Most likely, during the dry season
the condensation sink is too high for new particle formation.

**Acknowledgements**
We acknowledge the Academy of Finland Centre of Excellence program (grant no. 272041).
We acknowledge the Atmospheric Radiation Measurement (ARM) Climate Research Facility,
a user facility of the United States Department of Energy, Office of Science, sponsored by the
Office of Biological and Environmental Research, and support from the Atmospheric System
Research (ASR) program of that office. D.W. wishes to acknowledge the Austrian Science
Fund (FWF, grant no J-3951). P Artaxo acknowledges FAPESP project 2013/05014-0 and
CNPq for funding. We thank field support from Alcides Ribeiro, Bruno Takeshi and Fabio
Jorge. We acknowledge logistical support from the LBA Central Office, at the INPA – Instituto
Nacional de Pesquisas da Amazonia.

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

**Tables**
Table 1. A comparison between the sites during wet and dry season. Outside canopy site values
on the left, inside canopy on the right. The months chosen for the wet season for inside the
canopy are Jan-Mar and Dec-Mar for inside the canopy. Dry season includes Aug-Oct and
July-Sept for outside and inside canopy. Aerosol and ion parameters from the NAIS
measurements listed are ion concentrations in three size bins (0.8-2, 2-4 and 4-20 nm). Neutral
particle concentrations in two different size bins from the NAIS (2-4 and 4-20 nm) and total
particle concentrations (>10 nm) from CPC measurements and condensation sink from the
SMPS. The numbers present diel medians and in brackets 25th-75th percentiles. Environmental
parameters are temperature, relative humidity, precipitation, wind direction.




| | Outside canopy (T3) | | Inside canopy (T0t) | |
|---|---|---|---|---|
| **Particle and ion concentrations** | | | | |
| | *Wet* | *Dry* | *Wet* | *Dry* |
| **Cluster ions (0.8-2 nm) [cm⁻³]** | 1000 (-) (836-1500) | 988 (-) (688-1400) | 814 (-) (641-1051) 968(+) (790-1178) | 605(-) (465-801) 765(+) (604-1003) |
| **Intermediate ions (2-4 nm) [cm⁻³]** | 7 (-) (3-14) | 8 (-) (4-16) | 5(-) (2-10) 11(+) (7-17) | 4(-) (2-8) 11(+) (7-16) |
| **Large ions (4-20 nm) [cm⁻³]** | 58 (-) (27-107) | 56 (-) (30-106) | 84(-) (40-178) 147(+) (62-410) | 132(-) (52-425) 162 (80-329) |
| **Intermediate particles (2-4 nm) [cm⁻³]** | 579 (286-943) | 550 (276-927) | 477 (252-810) | 591 (323-1003) |
| **Large particles (4-20 nm) [cm⁻³]** | 1000 (547-2150) | 922 (552-1600) | 308 (169-690) | 530(-) (250-1070) |
| **CPC total particles (>10 nm) [cm⁻³]** | 1000 (533-1352) | 731 (411-2000) | - | - |
| **SMPS condensation sink [s⁻¹]** | 1.5 e-3 | 5 e-3 | - | - |
| **Environmental parameters** | | | | |
| **Temp [°C]** | 25.7 | 26.1 | 24 (23-25) | 24 (23-26) |
| **RH [%]** | 94.8 | 92.8 | 97 (93-98) | 96 (90-98) |
| **Precipitation [mm hr⁻¹]** | 49 | 32 | 0.7186 | 0.4248 |
| **Wind direction [°; relative to north]** | 92.6 | 112.7 | 97 (58-143) | 97 (60-147) |






Table 2. Annual statistics for ion and total particle concentration in T0t inside canopy site for
the period of 2011-2014. Values represent median (25th-75th percentiles).

| Size bin | Negative ion | Positive ions | Total particles |
|---|---|---|---|
| **0.8-2 nm** | 723 (537-1012) | 879 (683-1124) | - |
| **2-4 nm** | 5 (2-9) | 10 (7-16) | 521 (278-889) |
| **4-20 nm** | 85 (40-182) | 151 (68-382) | 380 (192-872) |


Table 3. New particle formation (NPF) characteristics at the outside canopy (T3) site.
Classified NPF event and rain-induced ion event frequencies.

|  | NPF days | Undefined | Non-events | Rain events | No-rain events |
|---|---|---|---|---|---|
| **Wet season (Jan-Mar)** | 8/65 (12%) | 0/65 (0%) | 57/65 (88%) | 61/65 (94%) | 04/65 (6%) |
| **Dry season (Aug-Oct)** | 0/49 (0%) | 0/49 (0%) | 49/49 (100%) | 15/49 (31%) | 34/49 (69%) |




Table 4. The parameters shown are from the outside canopy site for nucleation/no nucleation
event days. Median total particle concentration measured by a CPC, measured by the NAIS in
two size ranges (2-4 and 4-20 nm) and negative ion concentrations from the NAIS in three size
ranges (0.8-2, 2-4 and 4-20). The median values are calculated for the time window 09:00 –
12:00 as this is the time window of NPF events. The numbers in the brackets represent the 25th
and 75th percentile. The second part of the table includes median numbers of environmental
parameters for the whole day: temperature, RH, Precipitation and wind direction for NPF /no
NPF days. The main differences are the condensation sink and the wind direction.

| Particle and ion concentrations 09:00 – 12:00 LT | | |
|---|---|---|
| | **NPF day** | **Non NPF day** |
| **Cluster ions (0.8-2 nm) [cm⁻³]** | 800 (-) (692-905) | 870 (-) (687-1000) |
| **Intermediate ions (2-4 nm) [cm⁻³]** | 13 (-) (6-23) | 8 (-) (4-15) |
| **Large ions (4-20 nm) [cm⁻³]** | 83 (-) (44-137) | 62 (-) (25-119) |
| **Intermediate particles (2-4 nm) [cm⁻³]** | 606 (303-969) | 547 (522-1600) |
| **Large particles (4-20 nm) [cm⁻³]** | 1000 (604-1600) | 970 (238-1000) |
| **Full day data** | | |
| **SMPS Condensation sink [s⁻¹]** | 1e-3 | 5e-3 |
| **CPC total particles (>10 nm) [cm⁻³]** | 1100 (579-1860) | 1000 (404-2000) |
| **Environmental parameters full day** | | |
| | **NPF day** | **Non NPF day** |
| **Temp [°C]** | 26.5 | 25.9 |
| **RH [%]** | 90.6 | 93.6 |
| **Precipitation [mm hr⁻¹]** | 0 | 0.002 |
| **Wind direction [°; relative to north]** | 81.6 | 105.8 |






Table 5. Growth rates (GR, nmh$^{-1}$) and nucleation rates (J; cm$^{-3}$s$^{-1}$) determined from the NAIS
ion and particle data for each nucleation event. Also the median values for the condensation
sink (CS; s$^{-1}$) for each event day are shown. The GR were determined by finding the maximum
concentration for different size bins for 2-3 nm, 3-7nm and 7-20 nm. The GR values present
median values of the GR for positive and negative ions. The nucleation rates represent median
numbers for positive and negative ions. The GR between ions and particles agree well with
each other for the individual nucleation events. GR are smaller for the smaller size ranges and
increase with particle size. The nucleation rates are in general higher for the particles than the
ions. The Condensation is calculated from the SMPS size distributions.

| Size bins | 2-3 nm | | | | 3-7 nm | | | | | | |
|---|---|---|---|---|---|---|---|---|---|---|---|
| | *Particles* | | *ions* | | *Particles* | | *Ions* | | *Particles* | | |
| | GR (nm h$^{-1}$) | J (cm$^{-3}$s$^{-1}$) | GR (nm h$^{-1}$) | J (cm$^{-3}$s$^{-1}$) | GR (nm h$^{-1}$) | J (cm$^{-3}$s$^{-1}$) | GR (nm h$^{-1}$) | J (cm$^{-3}$s$^{-1}$) | GR (nm h$^{-1}$) | J (cm$^{-3}$s$^{-1}$) | CS (s$^{-1}$) |
| 29.01.2014 | 0.8 | 0.19 | 1.4 | 0.003 | 2.8 | 0.097 | 1.7 | 0.001 | - | - | - |
| 30.01.2014 | - | - | 3.7 | 0.011 | 13.6 | - | 7.1 | 0.13 | 4.4 | 0.38 | 0.0007 |
| 06.02.2014 | 0.8 | - | 19.8 | 0.07 | 29 | 0.87 | 11 | 0.01 | 24 | 0.28 | 0.0016 |
| 12.02.2014 | 0.7 | 0.17 | 1.2 | 0.005 | 1.3 | 0.09 | 1.2 | 0.003 | 3 | 0.09 | 0.0016 |
| 12.03.2014 | 1.1 | 0.2 | 1.7 | 0.002 | 13.3 | - | 11.2 | 0.008 | 4.9 | - | 0.0014 |
| 13.03.2014 | 1.5 | 0.2 | 1.6 | - | 1.2 | - | 8 | - | 13.6 | 0.24 | 0.0015 |
| 18.03.2014 | - | - | 0.7 | 0.002 | - | - | 7.7 | 0.009 | - | - | 0.0017 |
| 25.03.2014 | 0.8 | 0.11 | - | - | 15.7 | 0.4 | 15.8 | 0.018 | 14 | 0.18 | 0.0017 |





**Figures**

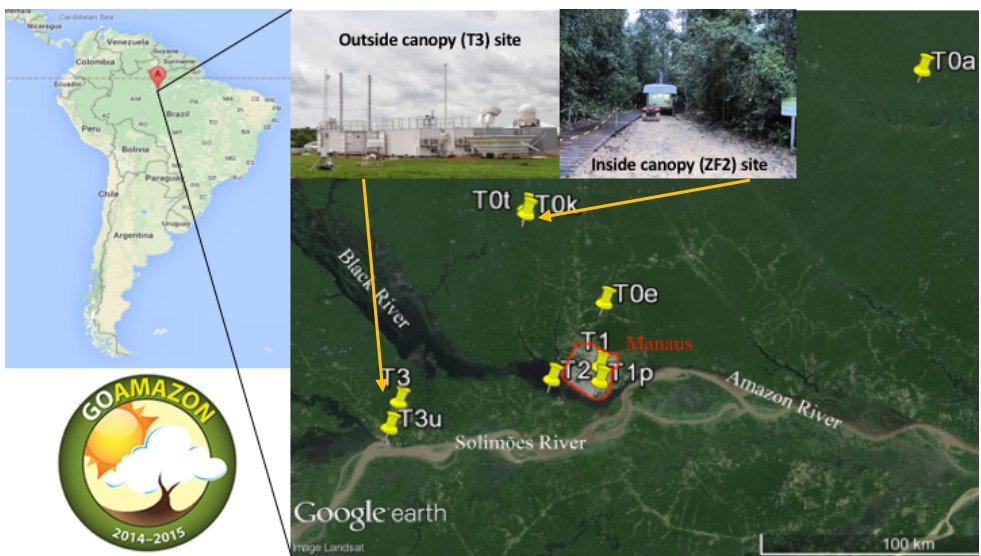

Figure 1. Location on map and photos of the inside canopy (T0t) and outside canopy (T3) sampling sites in Amazonas. The left column shows a map of South America and the right side shows a satellite view and photos of the T0t and T3 environment. From the inside canopy site we present the long term data, whereas from the outside canopy site we show the comparison of wet and dry season. The inside canopy measurements at T0t is located in a pristine area, whereas the outside canopy site is located at T3, downwind of Manaus.



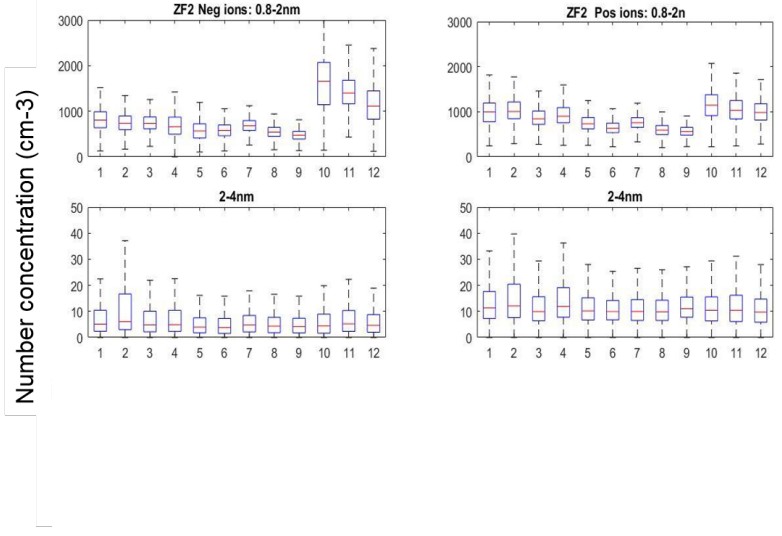

Figure 2. Annual variation of the negative (left column) and positive (right column) ion concentrations for 2011–2014 from inside the rainforest canopy. The bars represent median monthly ion concentrations, and the whiskers represent $25^{th}$ and $75^{th}$ percentiles.

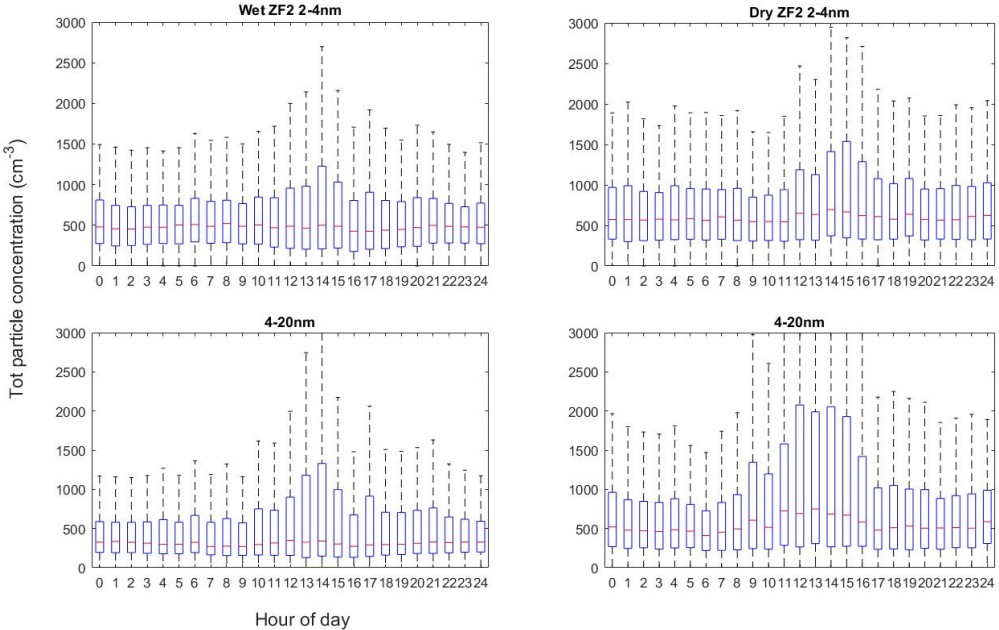



Figure 3. Diel cycle of total particle concentration at the inside canopy site (T0t) during the wet (Dec-Mar; left) and dry (Apr-Oct; right) season, for 2-4 nm (top) and 4-20 nm (bottom) particles.

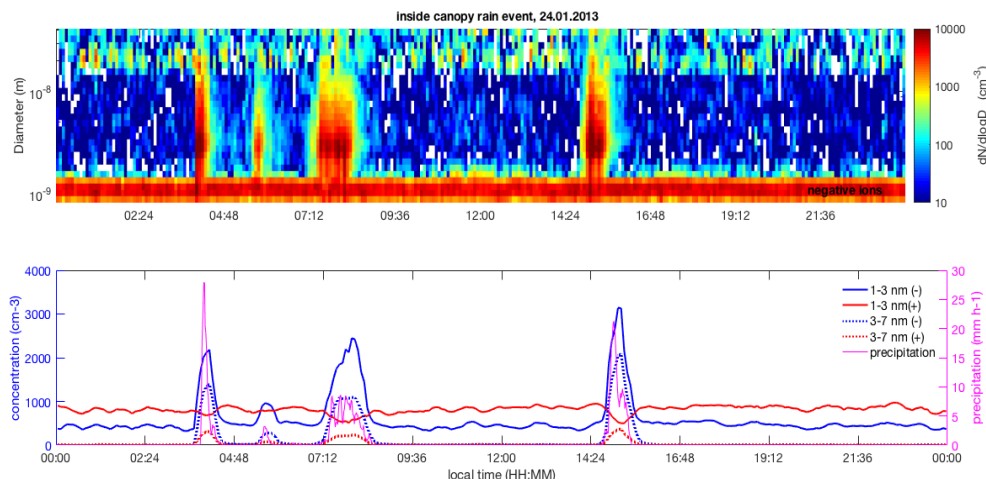

Figure 4. Typical rain-induced event at the inside canopy (T0t) site (Jan 24, 2013). Upper panel: negative ion number size distribution during a selected rain-induced ion production day measured with the NAIS. Lower panel: precipitation in mm h$^{-1}$ (pink trace) on the right axis, concentration of small (1-3 nm) negative (blue) and positive (red) ions with the scale on the left axis.





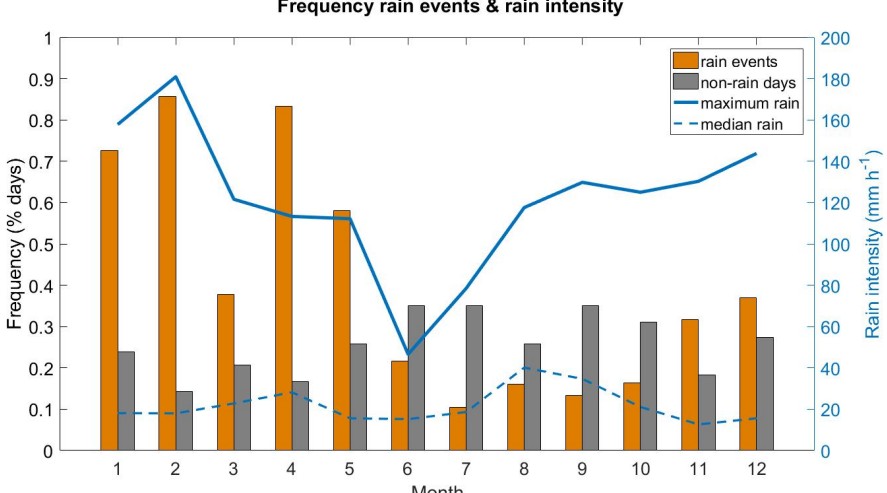

Figure 5. Inside canopy site (T0t) monthly frequencies (% of days with available data) of rain-induced ion events (n=306) and no-rain days (n=195). Dashed line indicates the median monthly variation of rain intensity (mm h$^{-1}$). Data collected continuously from September 2011 to January 2014 at the T0t site.

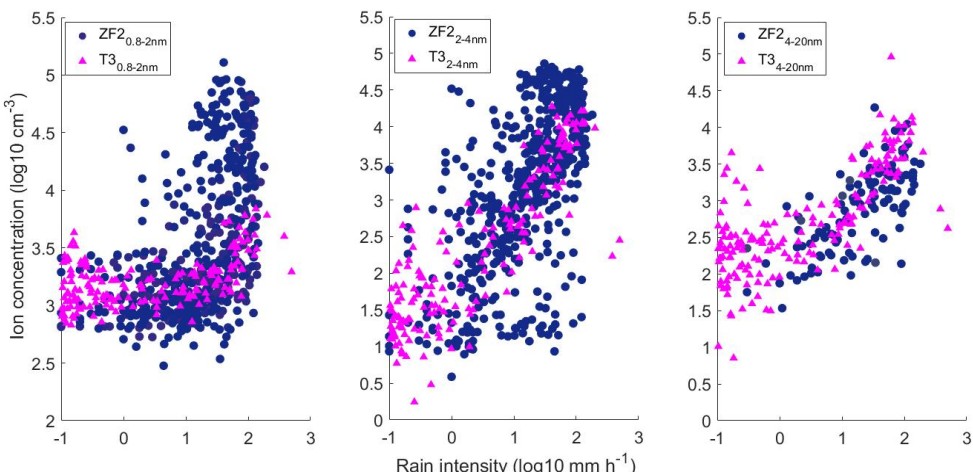

Figure 6. Median daily ion concentrations as a function of rain intensity at the inside canopy site (T0t) between September 2011 and January 2014 (blue circles). Outside canopy site (T3) rain events concentrations are added for comparison (triangles). (A) Cluster ions (0.8-2 nm), (B) 2-4 nm, and (C) 4-20 nm.



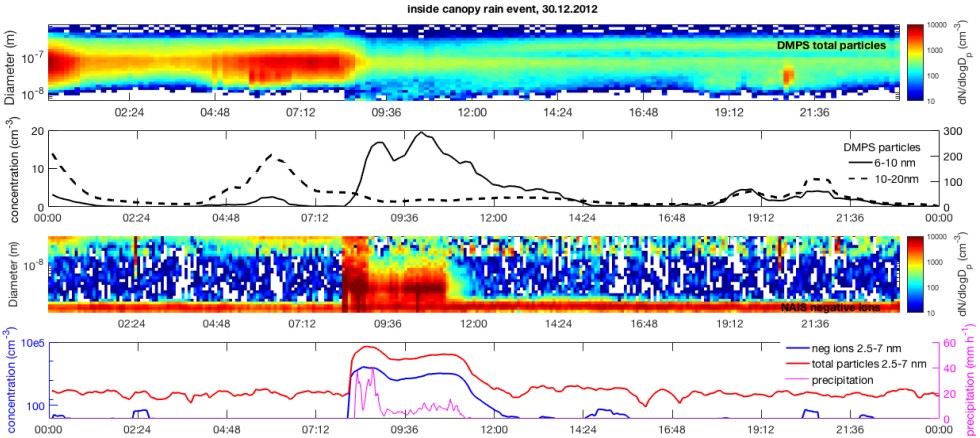

Figure 7. Example for a rain-induced event for total particles (DMPS). The DMPS measurements are taken above the canopy (60 m height), NAIS measurements are inside the canopy. Panel (a) shows the DMPS surface Figure. Panel (b) shows the particles measured by the DMPS for 6-10 nm and 10-20 nm (dashed line). (c) shows the surface Figure for the negative ions, measured by the NAIS. (d) shows the negative ion concentrations for 2.5-7 nm in blue and the total particle concentration in the same size range from the NAIS in red with the scale on the left axis. The pink trace shows the precipitation in mm h$^{-1}$ on the right axis.





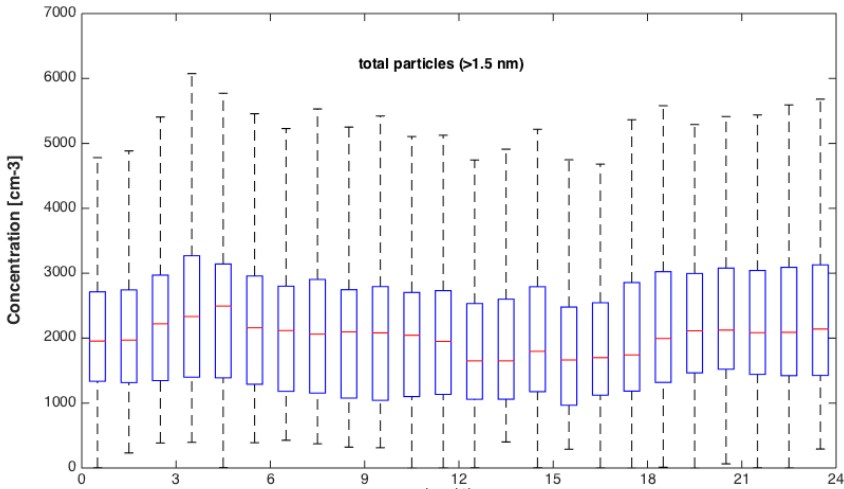

Fig 8. Diel cycle of particles bigger than 1.5 nm measured by the PSM during the dry season outside the rainforest canopy. In total, 38 days of data were used. The data shows hourly median concentrations, the whiskers 25[th] and 75[th] percentile.





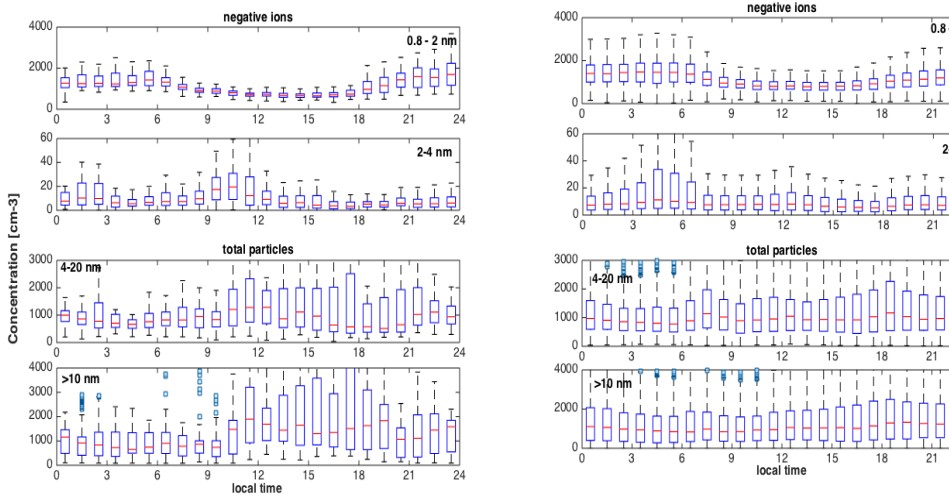

Figure 9. Diel cycle of aerosol particles and ions measured outside the canopy by the NAIS (small: 0.8–2 nm; intermediate: 2–4 nm; large: 4–20 nm) and total particles >10 nm as measured by the MAOS CPC. The left column shows the NPF event days and the right column the non NPF days. The markers are hourly median number concentrations and the whiskers 25th and 75th percentiles.





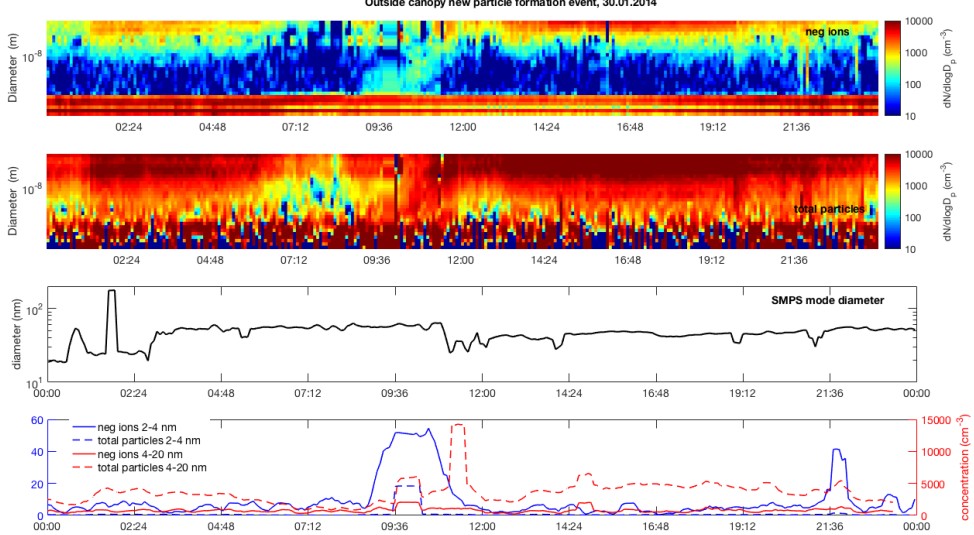

Figure 10. One example NPF day as observed at the outside canopy (T3) site. (a) and (b) show the surface Figures from the NAIS, (a) for negative ions, (b) for total particles. The color code indicates the measured concentrations. Panel (c) shows the mode diameter as measured by the MAOS SMPS. The mode diameter decreases at the start of the NPF event followed by continuous growth up to about 60 nm. Panel (d) shows negative ion and neutral particle concentrations in two size ranges (2-4 nm and 4-20 nm). Note the left axis is for the 2-4 nm ion concentration and 2-4 nm particle concentrations, right 4-20 neutral particle and ion concentrations.