# Peer review of "Ground-based observation of clusters and nucleation mode particles in the Amazon"

_Atmospheric Chemistry and Physics, 2017_

## Referee Comment (RC1) · Anonymous Referee #1 · 14 Sep 2017

The presented paper shows particle and ion number concentrations measured at two sites inside and close to the Amazon rainforest. The authors contrast seasonal and diurnal variations of the two different measurement stations by comparing one long-term data set to an intensive operational period during GoAmazon 2014/5. Thereby, rain-related particle and ion bursts are observed below canopy at the near-pristine forest station. New particle formation events have not been observed at the near-pristine site but at a pasture site downwind Manaus.

The manuscript contains valuable and scientifically significant information. Nevertheless, there are some aspects which need further analysis, deeper discussion and clarification. Although the paper is well written, figures and tables are lacking

significant information and need a careful revision. Some text passages, identical to existing work, are not referenced correctly. I suggest a careful major revision before considering publication in ACP.

**General comments:**

The authors compare two different research stations in the vicinity of Manaus. T0t - a remote, near pristine station inside the rainforest of an ecological reserve north of Manaus (e.g., Martin et al., 2016) and T3 - a pasture site (size 2.5 x 2 km) downwind of Manaus outside the rainforest. The observations in this manuscript are divided in 'inside canopy' (T0t) and 'outside canopy' (T3) which suggests a direct comparability and a close connection to forest canopies at both sites. Since T3 is not located inside the rainforest but a pasture site, I suggest to clearly discuss this in the manuscript. Instead of dividing into 'inside' and 'outside canopy', the authors should consider to rephrase the notation for T3 to 'outside forest' or 'pasture site' to avoid confusion.

Here shown particle number concentrations (in particular for the nucleation mode size range) are (at least in the wet season) significantly higher compared to former observations in the (pristine) Amazon region (e.g., Martin et al., 2010a, Martin et al., 2010b, Zhou et al., 2002). On average, the particle number concentration in the 4-20 nm size range agrees very well with the total particle concentration (> 10 nm) measured by a CPC (tables 1, 4), suggesting, that the majority of all particles is in the nucleation mode size range. These contradictory findings are not discussed by the authors. More exact information are addressed in my specific comments.

There are specific sentences and complete text passages which are identical to Martin et al., 2016. Those text passages are either not referenced or not referenced correctly. I suggest to put these sentences in quotation marks or to rephrase the respective sentences. In both cases the original source has to be cited correctly. The

corresponding text passages are listed in my specific comments.

**Specific comments:**

There are specific sentences and complete text passages which are identical to Martin et al., 2016. The following list is not necessary complete. The authors should make sure that further text passages similar to other work are referenced correctly. I encourage to use the similarity report provided by the iThenticate plagiarism screening service.

Wimmer et al. (page 3, lines 86 and following):
"The regular synoptic changes between the wet and dry seasons offered an additional important scientific contrast to study aerosol dynamics."
Martin et al. (p. 4787):
"The regular synoptic changes between the wet and dry seasons offered an additional important scientific contrast."

Wimmer et al. (page 5, lines 135 and following):
"Under the day-to-day variability in the meteorology, both clean and polluted air masses, mixed to variable degrees, arrived at T3."
Martin et al. (p. 4786):
"Under the day-to-day variability in the meteorology, both clean and polluted air masses, mixed to variable degrees, arrived at T3."

Wimmer et al. (page 7, lines 210 and following):
"The particle population is in dynamic balance with the ecosystem and anthropogenic contributions (e.g. biomass burning; which produces them directly and indirectly) and the hydrological cycle (which removes them). In the wet season (December to March), the Manaus plume aside, the Amazon basin is one of the cleanest continental regions on Earth (Andreae, 2007; Martin et al., 2010). In the dry and transition season (April to

September), biomass burning emissions are prevalent throughout the basin. The most intense biomass burning and atmospheric perturbations take place at the southern and eastern edges of the forest..."
Martin et al. (p. 4787):
"In the wet season, the Manaus plume aside, the Amazon basin is one of the cleanest continental regions on Earth (Andreae, 2007; Martin et al., 2010a). The particle population is in dynamic balance with the ecosystem (which produces them directly and indirectly) and the hydrologic cycle (which removes them). In the dry season, biomass burning is prevalent throughout the basin. The most intense burning and atmospheric perturbations take place at the southern and eastern edges of the vast forest."

Wimmer et al. (page 3, lines 114 and following):
"Manaus is situated at the confluence of the Black River (Rio Negro) with the 115 Solimões river, which together form the Amazon river. The city is an isolated urban region with a population of more than 2 million people (IBGE, 2015; Martin et al., 2017) and is surrounded by 1500 km of forests in all directions."
Martin et al. (p. 4786):
"Manaus, situated at the confluence of the Black River (Rio Negro) with the Solimões river, which together form the Amazon river, is an isolated urban region of over 2 million people (IBGE, 2015)."

Page 4, lines 134:
The authors state that T3 is located in a pristine environment. According to e.g., Martin et al., 2016 T3 (time points three) is located downwind of the pollution in a pasture area. I suggest to not use 'pristine' in this context.

Page 4, lines 118:
"T0t is mostly unaffected by the Manaus pollution and is surrounded by dense rainforest. It allows the characterization of an almost completely undisturbed natural environment" - Did the authors filter for pollution affected periods? If so, what are the filter criteria?

Page 4, lines 124:
The introduced DMPS measurements are performed using an inlet line above canopy. Nevertheless, the section is called 'inside canopy measurements' which is confusing. I further wonder if there are any comparisons of the DMPS and NAIS during the 3-year period to confirm the quality of measurements.

Page 5, lines 136:
"The site is located in a clearing of the rainforest .." According to Martin et al., 2016 the site is located in a pasture area (2.5 x 2 km) outside the rainforest. I suggest to rephrase the text accordingly from 'outside canopy' to 'outside forest' or 'pasture site'.

Page 6, lines 180:
A description of the applied inlet system for the PSM would be interesting for future studies under high rh conditions.

Page 6, lines 183:
"Laboratory studies have shown that the RH affects the counting efficiency of the PSM drastically" - Please provide references.

Page 7, lines 203:
"The DMPS data reported here is qualitative but not quantitative." - Please specify if there were problems with this instrument. Quantitative SMPS data are discussed in e.g., section 3.2.

Page 7, lines 220:
The planetary boundary layer development is probably different for pasture and rainforest sites. Can you please comment on that?

Page 8, lines 234:
"We observed an unexplained increase in the concentrations of the cluster ions in the NAIS towards the end of October 2013 to January" - Can you please comment on possible reasons for that drift? Is it possible that this drift continued after moving to T3?

Page 9, lines 276:
"the biomass burning during the dry season is expected to increase large ion concentrations" - Please provide a reference

Page 10, lines 287:
"Figure 2 shows the seasonal variability of ions and particles in the three size ranges (0.8-2nm, 2-4 nm and 4-20 nm)" - the lowermost panel in Fig. 2 is missing.

Page 10, lines 305:
In this paragraph it is not clear to which figure or table the authors refer to. Some examples:
"Positive and negative cluster ion concentrations were, on average, higher during the wet season compared to the dry season."
"Additionally, cluster ions (0.8-2 nm) showed slightly higher concentrations in the morning and evening, compared to other times of the day"
"A dip in the median ion concentration after midday coincides with a higher median concentration of large ions, which is a sign of a larger sink for cluster ions."
"Lastly, 4-20 nm ions peaked at around midday during the wet season, while their diel pattern was more irregular during the dry season."

Page 11, lines 343:
"The median total particle concentrations were about a factor of two higher during dry season (about 1500 cm-3) compared with the wet season (about 700 cm-3)." - In table 1 different values are shown. Furthermore, large particle (4-20 nm) concentrations are very similar to CPC measurements (> 10 nm), implying that on average all particles are in the size range between 10 and 20 nm.
Also, the average particle concentrations (4-20 nm) at T0t (250-800, for the wet season) compares well to total particle concentrations (e.g., in 10-500 nm size range) reported in earlier studies (e.g., Martin et al., 2010a, Martin et al., 2010b, Zhou et al., 2002). This again implies that the size distribution is dominated by nucleation mode particles, which is in contrast to the same mentioned references.

I kindly ask the authors to critically discuss these discrepancies.

Page 12, lines 361:
"The rain events were more common during the wet season (Fig. 5) when also the median rain intensity was higher." According to Fig 5, the median rain intensity is highest in August.

Page 11, lines 377 and following:
In section 3.2 the authors describe a very interesting and scientifically significant phenomenom of increased particle and ion concentrations during rain. Concentrations increase by 2 orders of magnitude towards more than 10000 particles/ions per cubic centimeter. In the following discussion, the authors mention that the particle concentration (nucleation mode size) above canopy (SMPS) does not increase accordingly. Instead, particle concentration increases only by 20 particles per cubic centimeter (6-20 nm size range), strongly contrasting the conditions below. They conclude that the high particle/ion concentration is a below canopy phenomenom. Furthermore, these nucleation mode particles are not able to leave the canopy which is acting as an umbrella preventing mixing.
In contrast, the presented diurnal variation suggests that mixing and planetary boundary layer development is efficient (although less efficient as compared to the pasture site). Also, the authors argue that they are able to measure ions and particles related to transported biomass burning plumes (page 9, lines 275). Why are those particles able to be mixed into the canopy. It is hard to believe that the forest canopy can maintain such a strong gradient of particle number concentration.
Please justify your statement.

Page 15, lines 454:
Please consider to show the results of your backward trajectory analysis in a map.

Page 15, lines 459:
In Fig. 10 a new particle formation event is shown: Please consider to add SMPS

contour plots and SMPS particle number concentrations in the nucleation mode size range. Statistical information of SMPS nucleation mode particle number concentration will add further valuable information to Figure 9 and Tables 1 and 4.

The absence of the forest canopy at T3 gives the opportunity to combine NAIS and SMPS measurements, which allows to investigate the entire evolution of the sub-micron aerosol population.

Page 17, lines 510:
"Similar, but weaker, rain-events were found at the site outside the rainforest canopy (T3)." - weaker in terms of what?

**Technical comments:**

Geolocations are indicated by e.g., -2° S, -60° W. South and west already indicate the negative latitude and longitude.

Page 2, line 72: "Martin et al., 2010" - a or b?

Page 7, line 202, 214, 225: "Martin et al., 2010" - a or b?

Page 9, lines 276: "Martin et al., 2010" - a or b?

Page 17, lines 531: "Amzon"

**Technical comments related to figures:**

In all boxplots, the whiskers are related to the 25th and 75th percentile. If this is true, what are the boxes referring to?

Fig 1:

- abbreviation "ZF2" used but not explained

Fig 2:

- abbreviation "ZF2" used but not explained

- lowermost 2 panels are missing

- "cm-3", "-3" has to be a superscript

Fig 3:

- "Tot part." - avoid unexplained abbreviations

- lower whiskers are all smaller than or equal zero, the authors state that the data are carefully cleaned. Please comment on the the large amount of zero and below zero particle concentrations.

- Is this the ion concentration in the upper panel and the total particle concentration in the lower panel? The caption and axis only refers to particles.

Fig 4:

- time axis of different panels not synchronized

- upper panel: tick marks invisible

- cm-3, mm h-1

- if positive and negative ions share the same axis, it should not be blue

- number concentration of small positive and negative ions disagrees by a factor of 2. According to Manninen et al., 2016 there should be an agreement within 20%. Please comment on the instrument performance and data quality.
Fig 5:

- please check if the left axis label "%" is correct

- please consider to also show the average total rain

Fig 7:

- time axis of different panels not synchronized

- tick marks in contour plots invisible

- second panel shows additional axis on right side

- lowest panel shows more than 2 orders of magnitude with just two labels

Fig 8:

- "cm-3"

Fig 9:

- please consider to label the 2 columns with NPF  no NPF

- the individual panels are irregularly distributed

**Technical comments related to tables:**

Tab 1:

- almost all parameters show IQR, please add missing IQR values for CS and meteorological parameters

- precipitation rate is given with 4 digits, does that represent the measurement accuracy?

- please consider to add total precipitation and wind velocity

Tab 4:

- NAIS size range indicated in caption misses units

- again, please add IQR for all parameters

- please consider to add total precipitation and wind velocity

Tab 5:

- the table does not fit the page, last column is not readable

- the size information in last columns is missing

**Literature:**

Manninen, H. E., Mirme, S., Mirme, A., Petäjä, T. and Kulmala, M.: How to reliably detect molecular clusters and nucleation mode particles with Neutral cluster and Air Ion Spectrometer (NAIS), Atmos. Meas. Tech., 9(8), 3577–3605, doi:10.5194/amt-9-3577-2016, 2016.

Martin, S. T., Andreae, M. O., Althausen, D., Artaxo, P., Baars, H., Borrmann, S., Chen, Q., Farmer, D. K., Guenther, A., Gunthe, S. S., Jimenez, J. L., Karl, T., Longo, K., Manzi, A., Müller, T., Pauliquevis, T., Petters, M. D., Prenni, A. J., Pöschl, U., Rizzo, L. V., Schneider, J., Smith, J. N., Swietlicki, E., Tota, J., Wang, J., Wiedensohler, A. and Zorn, S. R.: An overview of the Amazonian Aerosol Characterization Experiment 2008

(AMAZE-08), Atmos. Chem. Phys., 10(23), 11415–11438, doi:10.5194/acp-10-11415-2010, 2010a.

Martin, S. T., Andreae, M. O., Artaxo, P., Baumgardner, D., Chen, Q., Goldstein, A. H., Guenther, A., Heald, C. L., Mayol-Bracero, O. L., McMurry, P. H., Pauliquevis, T., Pschl, U., Prather, K. A., Roberts, G. C., Saleska, S. R., Silva Dias, M. A., Spracklen, D. V., Swietlicki, E. and Trebs, I.: Sources and properties of Amazonian aerosol particles, Rev. Geophys., 48(2), RG2002, doi:10.1029/2008RG000280, 2010b.

Martin, S. T., Artaxo, P., MacHado, L. A. T., Manzi, A. O., Souza, R. A. F., Schumacher, C., Wang, J., Andreae, M. O., Barbosa, H. M. J., Fan, J., Fisch, G., Goldstein, A. H., Guenther, A., Jimenez, J. L., Pöschl, U., Silva Dias, M. A., Smith, J. N. and Wendisch, M.: Introduction: Observations and Modeling of the Green Ocean Amazon (GoAmazon2014/5), Atmos. Chem. Phys., 16(8), 4785–4797, doi:10.5194/acp-16-4785-2016, 2016.

Zhou, J., Swietlicki, E., Hansson, H. C. and Artaxo, P.: Submicrometer aerosol particle size distribution and hygroscopic growth measured in the Amazon rain forest during the wet season, J. Geophys. Res. D Atmos., 107(20), doi:10.1029/2000JD000203, 2002.
* * *

---

## Referee Comment (RC2) · Anonymous Referee #2 · 24 Oct 2017

This work presents ground-based particle ion and number concentration measurements from the Amazon. The results are interesting and useful, but the paper needs to be carefully edited before it can be considered for publication in ACP. Also, as discussed in comment 18 below, I think the authors need to consider an additional explanation for their below vs above canopy rain-induced example that is illustrated in Figure 7.

Specific comments:

1) Title: "Direct" seems unnecessary. Perhaps better replaced by "Ground-based"? Also, "molecular clusters" seems inappropriate. Perhaps "particle ions"?

2) Line 38 – "Pristine" is used here and in a few other places. It needs to be defined.

[Figure]

3) Lines 40-42 - Define the sites as locations relative to Manaus, much as you did on lines 80-83. You can't expect all readers to identify with T0t and T3.

4) Lines 43-44 – "T0t is reached by the pollution about 1 day in 7, where the T3 site is about 15% of the time affected by Manaus." The statement implies a difference between T0t and T3, but 1 in 7 is 14%, which is not different from 15%. What are you trying to say here?

5) Lines 59-60 – This sentence is not useful. Also, you state in the paper that the back trajectories in both cases pass over Manaus. Does not the source strength of Manaus even out other differences in the trajectories? Your last sentence of the conclusions is that "Most likely, during the dry season the condensation sink is too high for new particle formation." That appears to be the main factor that differentiates between the NPF and non-NPF days. Why is that not mentioned in the abstract?

6) Lines 221-222 – You say "The vertical mixing can be enhanced during the wet season due to convective clouds." Are you saying that convective clouds lift the mixed layer or that convective clouds lift particles out of the mixed layer or something else? Clouds formed at the top of a mixed layer will tend to cool below, which does not help the development of a mixed layer.

7) Lines 275-279 – This may be true for inside the canopy, but not for outside the canopy. Please clarify. Also, why would the pattern outside of the canopy not reflect biomass burning and wet deposition more than that inside the canopy?

8) Lines 281-282 - That appears to be true for the wet season, but the factor is less than 2 during the dry season. Did you mean "up to a factor of 3"?

9) Lines 287-288 – The 4-20 nm ions are not shown in the Figure 2 I have.

10) Line 301 – "Oct-Dec for both seasons"? Oct-Dec is a season (fall). Specify wet and dry seasons.

11) Line 305 – On line 214 the dry and transition season is April to September, whereas

here it is Apr-Oct. Please correct.

12) Line 311-312 – Cluster ions are not shown in Figure 3. Where are we supposed to view this?

13) Lines 321-323 – Again, 4-20 nm ions are not shown in Figure 2.

14) Line 358 – What do you mean when you say that "negative ions are smaller than positive ions"? Do you mean fewer in number?

15) Lines 374-376 and figure 6 – For the ions in the 0.8-2 nm particles, it looks like they simply turn on at rain intensities above 1.

16) Figure 7 – Indicate which axis corresponds with which particle size class in Panel B; presumably, the LH axis is 6-10 nm.

17) Line 379 – "followed by a second one at about 11:00". Here, indicate the relative difference in rain intensity.

18) Figure 7 and lines 385-395 – This is a very interesting set of observations. If particles descending with the rain were responsible for the increase in 6-10 nm particles above the canopy, how do you explain the apparent evolution of 6-10 nm particles to 10-20 nm over a few hours? Given the roughly 3 orders of magnitude difference in particle number concentrations from ground to above canopy and the potential canopy filtering you mention, why instead is it not possible that the few 6-20 nm particles above the canopy were due to the rain-induced particles mixing and filtering upwards?

19) Table 3 and lines 404-406 - Table 3 shows 65 and 49 for a total of 114, while you state 64 and 46 and 113. Please correct.

20) Figure 9 – On either side, you show four panels. The top two are labelled ions and the bottom two are labelled total particles, which is consistent with the text. In the caption, we are led to believe that the top three are ions. Please correct.

21) The RH side of Table 5 is cut off in my copy.

22) Line 520 – Should be Jan-March for wet season?

23) A couple of more general comments: Is there some sort of summary connecting the ion concentrations with NPF that can be drawn? The rain-induced events are prominent, but we are not given any sense of how important these might be. For example, is there any evidence that a significant number of rain-induced particles survive to become CCN size, or is Figure 7 the best example of their potential longevity?
* * *

---

## Author Comment (AC1) · 13 Dec 2017

Replies to referee #1

We thank the referee for the careful revision of the manuscript 'Direct observations of molecular clusters and nucleation mode particles in the Amazon'.

The comments improve the current manuscript. We will address all the comments and concerns in detail as shown below/ as in the following paragraphs.

General comments. We thank the reviewer for suggesting the comparison of the two research sites. This issue has been addressed carefully in the revised manuscript.

The identity of specific sentences in the current manuscript were a mistake. We

have re-phrased the identical sentences from previous publications in the revised manuscript.

We address the specific comments of the referee here below.

Referee comment: There are specific sentences and complete text passages which are identical to Martin et al., 2016. The following list is not necessary complete. The authors should make sure that further text passages similar to other work are referenced correctly. I encourage to use the similarity report provided by the iThenticate plagiarism screening service.

Reply: All the identical text passages to previous publications have been re-phrased in the revised manuscript.

Specific comments: Page 4, lines 134: The authors state that T3 is located in a pristine environment. According to e.g., Martin et al., 2016 T3 (time points three) is located downwind of the pollution in a pasture area. I suggest to not use 'pristine' in this context. Reply: We agree with the referee. The term pristine has been removed from the revised manuscript

Page 4, lines 118: "T0t is mostly unaffected by the Manaus pollution and is surrounded by dense rainforest. It allows the characterization of an almost completely undisturbed natural environment" - Did the authors filter for pollution affected periods? If so, what are the filter criteria? Reply: In the general data analysis, we did not filter for pollution affected periods, since we report average values for the whole measurement period and wet/dry season specifically. However, for the analysis of the NPF events, pollution events would appear in the NAIS/SMPS data as elevated aerosol concentrations in the accumulation mode. Also, the calculation of the condensation sink gives a good criterion for polluted days, which is clearly higher on non NPF days. Since we observed two nucleation events, with GR of approximately 10-20 nmh-1 and about 1 nmh-1, it might be that the days with the higher GR are days which are more influenced by the Manaus pollution plume. Since the sulfuric acid concentrations seemed to be about

the same on days with high and low GR, we may assume that the Manaus pollution is not the main factor influencing the air masses.

Page 4, lines 124: The introduced DMPS measurements are performed using an inlet line above canopy. Nevertheless, the section is called 'inside canopy measurements' which is confusing. I further wonder if there are any comparisons of the DMPS and NAIS during the 3-year period to confirm the quality of measurements. Reply: The section has been re-named to 'Measurements inside the rainforest' to avoid confusion. We changed the classification of the two sites in the whole manuscript accordingly. T0t is called inside rainforest site and T3 pasture or outside rainforest site. The instrumentation was calibrated before shipping to the campaign and regular maintenance including flow adjustments and zero checks were performed.

Page 5, lines 136: "The site is located in a clearing of the rainforest." According to Martin et al., 2016 the site is located in a pasture area (2.5 x 2 km) outside the rainforest. I suggest to rephrase the text accordingly from 'outside canopy' to 'outside forest' or 'pasture site'. Reply: We thank the referee for the suggestion. The text has been rephrased accordingly, line 177-180: 'The site is an open pasture site, where the Manaus pollution plume regularly intersects and the rainforest canopy did not hinder mixing. Due to the site location, T3 is either a pristine environment or highly influenced by the Manaus pollution plume, mainly depending on the wind direction.'

Page 6, lines 180: A description of the applied inlet system for the PSM would be interesting for future studies under high rh conditions. Reply: We agree with the referee. A description has been added to the revised manuscript, line 236-243: The inlet system consists of a core sampling probe combined with a sintered tube. The core sampling probe consists of two cylindrical tubes with different outer diameters (10 mm and 6 mm). The larger diameter of the outer tube allows up to 10 Lpm total laminar flowrate, to minimize diffusional losses. The inner tube is directly attached to the PSM with an airflow of 2.5 Lpm. The excess airflow is discarded into an exhaust line (Kangasluoma et al, 2016). Downstream of the core sampling line is a sintered tube where dry pressurized air is introduced. The water molecules in the sample flow are pushed towards the outer walls of the sinter material by diffusion, drying the airflow.

Page 6, lines 183: "Laboratory studies have shown that the RH affects the counting efficiency of the PSM drastically" - Please provide references. Reply: the sentence has been rephrased as follows: Line 244-246: Laboratory studies have shown that the RH affects the counting efficiency of the PSM drastically (higher sensitivity at smaller sizes at higher RH; Kangasluoma et al. 2013, Iida et al, 2009).

Page 7, lines 203: "The DMPS data reported here is qualitative but not quantitative." - Please specify if there were problems with this instrument. Quantitative SMPS data are discussed in e.g., section 3.2. Reply: the issue is addressed more precisely in the revised manuscript. Since the particle losses in the sampling line due to diffusion are not precisely known, the SMPS data has not been corrected for those losses. Hence, for the data shown in Figure 7, where the concentrations of 6-10 nm and 10-20 nm are shown, we feel comfortable only at making assumptions based on the trend of the data but not absolute numbers. We added the following sentence to the revised manuscript: Page 7, lines 264-267: 'The DMPS data reported here are qualitative, not quantitative, as the losses due to diffusion in the sampling line are not precisely known and therefore not taken into account in the data presented later in this manuscript.'

Page 7, lines 220: The planetary boundary layer development is probably different for pasture and rainforest sites. Can you please comment on that? Reply: The local features and land-use affect the development of convective boundary layer as well as their emission spectra in terms of volatile organic compounds are different. In the morning, the boundary layer develops more rapidly in the pasture area due to lower evapotranspiration and the sensitive heat flux is dominating. This induces a more rapidly growing mixed layer, causing more efficient vertical mixing of precursors and aerosols. Also, photochemistry is more pronounced in (semi) open area than under the canopy. However, during the daytime the small-scale variability in boundary layer dynamics and in VOC concentrations tends to even out. The rapid oxidation chemistry remains characteristic for each site. We added a sentence in the revised manuscript. Page 8, line 293-297: 'The boundary layer development is also different at the two different measurement sites. It develops more rapidly in the pasture area, causing a more efficient vertical mixing compared to the site enclosed by rainforest. From our observations, we conclude that the main differences in the dynamics of the aerosol particle population at the two measurement sites is due to the 'umbrella effect' of the rainforest canopy. Page 8, lines 234: "We observed an unexplained increase in the concentrations of the cluster ions in the NAIS towards the end of October 2013 to January" - Can you please comment on possible reasons for that drift? Is it possible that this drift continued after moving to T3? Reply: we carefully looked at the flow rates and other NAIS technical data that could give some input, but we could not find any clear indicator of an instrumental drift. The drift continued after moving to T3, which is why we corrected all the data after we observed the drift for the first time accordingly. We attribute the drift is caused by a slow change in the differential mobility analyzer flow rates and charger ion filtration that cause erroneously some of the corona charger generated ions to penetrate into the detectors. We explain this with the following sentences on page 8, lines 306-323 in the revised manuscript: 'We observed an unexplained increase in the concentrations of the cluster ions in the NAIS towards the end of October 2013 to January 2014 at the T0t site. This increased level continued when the NAIS was taken to the T3 site. We consider this drift instrumental. By comparing the 2014 concentrations of the NAIS channels to those prior to the increase (January 2012 and 2013), a correction factor of 1.8 was applied to the 4 smallest size channels of the NAIS (0.8-1.25 nm) to account for the drift for the subsequent data.'

Page 9, lines 276: "the biomass burning during the dry season is expected to increase large ion concentrations" - Please provide a reference Reply: we rephrased the sentence as follows: Page 9, lines 361-365: 'Additionally, the wet and dry seasonality characteristic for the Amazon (Rissler et al. 2006, Martin et al. 2010a) can be observed in the concentration of the large ions (4-20nm): the biomass burning during the dry season seems to increase large ion concentrations, whereas during the wet

season their concentrations decreased, most likely due to wet deposition and reduced source strengths. Page 10, lines 287: "Figure 2 shows the seasonal variability of ions and particles in the three size ranges (0.8-2nm, 2-4 nm and 4-20 nm)" - the lowermost panel in Fig. 2 is missing. Reply: This is a mistake. During the writing process, we decided not to show the 4-20 nm size range as it does not add any additional valuable information. The sentence was changed in the revised manuscript as follows: Page 10, lines 373-374: 'Figure 2 shows the monthly variability of ions and particles in two size ranges (0.8-2nm, 2-4 nm) for the 2011-2014 period.'

Page 10, lines 305: In this paragraph it is not clear to which figure or table the authors refer to. Some examples: "Positive and negative cluster ion concentrations were, on average, higher during the wet season compared to the dry season." "Additionally, cluster ions (0.8-2 nm) showed slightly higher concentrations in the morning and evening, compared to other times of the day" "A dip in the median ion concentration after midday coincides with a higher median concentration of large ions, which is a sign of a larger sink for cluster ions." "Lastly, 4-20 nm ions peaked at around midday during the wet season, while their diel pattern was more irregular during the dry season." Reply: this paragraph has been deleted from the revised manuscript. The numbers refer to a Figure that has been removed from the final manuscript, as we decided to only show the particle concentrations, as the data shows a very similar behavior as the ion data. The ion data does not add any additional information to the manuscript.

Page 11, lines 343: "The median total particle concentrations were about a factor of two higher during dry season (about 1500 cm-3) compared with the wet season (about 700 cm-3)." - In table 1 different values are shown. Furthermore, large particle (4-20 nm) concentrations are very similar to CPC measurements (> 10 nm), implying that on average all particles are in the size range between 10 and 20 nm. Also, the average particle concentrations (4-20 nm) at T0t (250-800, for the wet season) compares well to total particle concentrations (e.g., in 10-500 nm size range) reported in earlier studies (e.g., Martin et al., 2010a, Martin et al., 2010b, Zhou et al., 2002). This again implies

that the size distribution is dominated by nucleation mode particles, which is in contrast to the same mentioned references. Reply: The numbers reported in the text are a mistake. The sentence has been re-phrased as follows (page 11, lines 426-428 in the revised manuscript): The median total particle concentrations were about a factor of 1.5 higher during wet season (about 1000 cm-3) compared with the dry season (about 700 cm-3). The presented manuscript is (to our knowledge) the first comprehensive study of small ions and particles in the Amazon basin. We agree with the referee that from looking at those numbers, we could conclude that the aerosol particle population in the Amazon is dominated by the nucleation mode. Nevertheless, we should be careful since previous studies have not been focusing on nucleation mode particles. All the numbers presented in the current manuscript for the T0t site are directly from the measurements with the NAIS. Whereas the previous results have been using different instrumentation and the measurement locations have been different. We think that from our current knowledge we cannot conclude that the aerosol particle population in the Amazon is dominated by the nucleation mode.

Page 12, lines 361: "The rain events were more common during the wet season (Fig. 5) when also the median rain intensity was higher." According to Fig 5, the median rain intensity is highest in August. Reply: The sentence has been rephrased as follows: p. 12, lines 482-483 in the revised manuscript: 'The rain events were more common during the wet season, peaking in August which can be considered as transition season (Fig. 5; Martin et al, 2010) when also the median rain intensity was higher.'

Page 11, lines 377 and following: In section 3.2 the authors describe a very interesting and scientifically significant phenomenom of increased particle and ion concentrations during rain. Concentrations increase by 2 orders of magnitute towards more than 10000 particles/ions per cubic centimeter. In the following discussion, the authors mention that the particle concentration (nucleation mode size) above canopy (SMPS) does not increase accordingly. Instead, particle concentration increases only by 20 particles per cubic centimeter (6- 20 nm size range), strongly contrasting the conditions below. They conclude that the high particle/ion concentration is a below canopy phenomenom. Furthermore, these nucleation mode particles are not able to leave the canopy which is acting as an umbrella preventing mixing. In contrast, the presented diurnal variation suggests that mixing and planetary boundary layer development is efficient (although less efficient as compared to the pasture site). Also, the authors argue that they are able to measure ions and particles related to transported biomass burning plumes (page 9, lines 275). Why are those particles able to be mixed into the canopy. It is hard to believe that the forest canopy can maintain such a strong gradient of particle number concentration. Please justify your statement. Reply: Earlier studies have shown that rain and particularly shattering of water droplets will result in high concentration of ions (e.g. Tammet et al., 2009). Typically, these effects are not seen with aerosol instruments as the ions are neutralized in the measurement process. Our main point here is that this increase in ion concentrations is mainly an effect that can be observed inside the canopy as the ions that we observe are produced by splashing of the water droplets on the tree leaves. Those ions will not survive until the measurements by the DMPS as it is sampling from above the canopy and they ions are filtered out by the leaves before reaching the inlet of the DMPS. From the current measurements, we cannot make any statement of the source of the larger neutral particles that are seen by the DMPS above the canopy. It is likely that they are produced in cloud outflow regions and due to strong downdrafts entrained back into the mixing layer (Wang et al, 2016). Most likely the increase of 4-20nm ions during the dry season is a combination of local biomass burning sources and a decrease in wet deposition. The sentence has been re-phrased in the revised manuscript: Page 9, lines 361-365: 'Additionally, the wet and dry seasonality characteristic for the Amazon (Rissler et al. 2006, Martin et al. 2010a) can be observed in the concentration of the large ions (4-20nm): the biomass burning during the dry season seems to increase large ion concentrations, whereas during the wet season their concentrations decreased, most likely due to wet deposition and reduced source strengths. Page 15, lines 45 Please consider to show the results of your backward trajectory analysis in a map. Reply: We thank the referee for the suggestion,

as it improves the manuscript. We have added a Figure showing the map to clarify the back trajectory calculations We also rephrased the sentence as follows, line 594-598: 'These air masses all originate from upstream of the Amazon river, where the NPF day air mass originate from further north, which is an area with dense rainforest. The results of the back trajectory calculations are shown in Figure 10. The red line shows the median of an ensemble or the non event days and the blue line for NPF days.

Page 15, lines 459: In Fig. 10 a new particle formation event is shown: Please consider to add SMPS contour plots and SMPS particle number concentrations in the nucleation mode size range. Statistical information of SMPS nucleation mode particle number concentration will add further valuable information to Figure 9 and Tables 1 and 4. The absence of the forest canopy at T3 gives the opportunity to combine NAIS and SMPS measurements, which allows to investigate the entire evolution of the submicron aerosol population. Reply: we thank the referee for this suggestion. The SMPS Figure has been added to the Figure. Page 17, lines 510: "Similar, but weaker, rain-events were found at the site outside the rainforest canopy (T3)." - weaker in terms of what? Reply: We have re-phrased the sentence as follows in the revised manuscript: Page 17, lines 679-683: 'Similar rain-events were found at the pasture site (T3). The production of small (0.8-2 nm) and intermediate ions (2-4 nm) during rain events reached a maximum of 104 cm-3 at the pasture site, where it was one order of magnitude higher at the T0t site. Large ion concentrations reached similar concentrations during rain events at both measurement sites.'

Technical comments related to Figures The boxes refer to the 25th-75th percentile. Reply: The whiskers show the extreme values of the data set which are not considered outliers.

The tables and Figures have been changed according to the suggestions of the referee in the revised manuscript.

Fig 4: number concentration of small positive and negative ions disagrees by a factor

of 2. According to Manninen et al., 2016 there should be an agreement within 20%. Please comment on the instrument performance and data quality.

Reply: Table 1 shows a very good agreement between the positive and negative ion concentrations. We believe that the difference seen in Figure 4 is due to a problem with the instrument performance, which might be different on certain days, but which does not affect the overall good instrument performance and data quality.

References Iida, K., Stolzenburg, M. R., and McMurry, P. H.: Effect of Working Fluid on Sub-2 nm Particle Detection with a Laminar Flow Ultrafine Condensation Particle Counter, Aerosol Sci. Tech., 43, 81–96, 2009.

Kangasluoma, J., Junninen, H., Lehtipalo, K., Mikkilä, J., Vanhanen, J., Attoui, M., Sipilä, M., Worsnop, D., Kulmala, M., and Petäjä, T.: Remarks on Ion Generation for CPC Detection Efficiency Studies in Sub-3-nm Size Range, Aerosol Sci. Tech., 47, 556–563, 2013.

Kangasluoma, J., Franchin, A., Duplissy, J., Ahonen, L., Korhonen, F., Attoui, M., Mikkilä, J., Lehtipalo, K., Vanhanen, J., Kulmala, M., Petäjä, T. (2016b). Operation of the Airmodus A11 nano Condensation Nucleus Counter at various inlet pressures, various operation temperatures and design of a new inlet system. Atmos Meas Tech 9:2977-2988. Tammet, H., Hõrrak, U., and Kulmala, M.: Negatively charged nanoparticles produced by splashing of water, Atmos. Chem. Phys., 9, 357-367, doi:10.5194/acp-9-357-2009, 2009. Wang J., Krejci R., Giangrande S., Kuang, C., Barbosa H. M. J., Brito J., Carbone S., Chi X., ComstockJ., Ditas F., Lavric J., Manninen H. E., Mei F., Moran-Zuloaga D., Pöhlker C., Pöhlker M. L., Saturno J., Schmid B., Souza R. A. F., Springston S. R., Tomlinson J. M., Toto T., Walter D., Wimmer D., Smith J. N., Kulmala M., Machado L.A. T., Artaxo P., Andreae M. O., Petäjä T. & Martin S. T., Amazon boundary layer aerosol concentration sustained by vertical transport during rainfall, Nature, doi:10.1038/nature19819, 2016.

Please also note the supplement to this comment:
https://www.atmos-chem-phys-discuss.net/acp-2017-782/acp-2017-782-AC1-supplement.pdf
* * *
[Figure]

[Figure]

**Fig. 1.** Figure 10: median back trajectories for NPF (blue) and non NPF (red) days. The trajectories were calculated 24hours backwards arriving at 09:00 local time at 500m a.s.l. at the measurement site.

[Figure]

**Fig. 2.** Figure 11 One example NPF day as observed at the outside canopy (T3) site. (a) shows
the surface Figure from the SMPS, (b) and (c) show the surface Figures from the NAIS, (b) for
negative ions, (c) fo

[Figure]

**Fig. 3.** Figure 7. Example for a rain-induced event for total particles (DMPS). The DMPS measurements are taken above the canopy (60 m height), NAIS measurements are inside the canopy. Panel (a) shows the DMPS

---

## Author Comment (AC2) · 13 Dec 2017

Referee comment: 1) Title: "Direct" seems unnecessary. Perhaps better replaced by "Ground-based"? Also, "molecular clusters" seems inappropriate. Perhaps "particle ions"? Reply: we thank the referee for the valuable suggestion for the title. We agree that "Ground-based" suits the content of the paper better than "direct". We would like to keep "molecular clusters though, since with the instrumentation used in our work, we are able to measure down to the molecular cluster level.

Referee comment 2) Line 38 – "Pristine" is used here and in a few other places. It needs to be defined. Reply: we agree with the reviewer that an explanation of pristine in our manuscript is missing. We added the following sentences in the revised manuscript:

[Figure]

Line 38 and following; 'The occurrence of NPF on ground level in the Amazon region has not been observed previously in pristine conditions. In this work, pristine refers to CCN concentrations of a few hundred cm-3.'

Referee comment 3) Lines 40-42 - Define the sites as locations relative to Manaus, much as you did on lines 80-83. You can't expect all readers to identify with T0t and T3. Reply: We agree with the referee that a description of the measurement locations is missing at this point of the manuscript. We changed the sentence in the revised manuscript as follows, line 40-45: We measured the variability of air ion concentrations (0.8–20 nm) with an ion spectrometer between 2011 and 2014 at the T0t site and between February and October 2014 at the GoAmazon 2014/5 T3 site. The T0t site is surrounded by dense rainforest, mostly unaffected by the Manaus pollution plume. The T3 site, instead is an open pasture site, 70km downwind of Manaus.

Referee comment 4) Lines 43-44 – "T0t is reached by the pollution about 1 day in 7, where the T3 site is about 15% of the time affected by Manaus." The statement implies a difference between T0t and T3, but 1 in 7 is 14%, which is not different from 15%. What are you trying to say here? Reply: We agree with the referee that here the numbers are similar and the sentence is confusing. We were trying to point out the differences between the two measurement sites. The T0t site is parallel wind to the Manaus pollution plume, where the T3 site is downwind of the Manaus pollution plume. We discussed the numbers again. Based on AMAZE-08, we concluded T0t is affected about once a week (Martin et al. 2010 (in Supp Material) T3 gets influenced between once every day and once every two days for a few hours, especially in the afternoon (de Sa et al. 2017, Thalman et al. 2017). We rephrased the sentence to, line 47-50: 'T0t is influenced by pollution about once per week, where T3 on the other hand is reached once per day/once per every second day, especially in the afternoon (Martin et al., 2010b supplementary material, Thalmann et al, 2017, de Sa et al, 2017).'

Referee comment 5) Lines 59-60 – This sentence is not useful. Also, you state in the paper that the back trajectories in both cases pass over Manaus. Does not the source

strength of Manaus even out other differences in the trajectories? Your last sentence of the conclusions is that "Most likely, during the dry season the condensation sink is too high for new particle formation." That appears to be the main factor that differentiates between the NPF and non-NPF days. Why is that not mentioned in the abstract? Reply: We agree with the referee that the difference in the condensation sink between NPF and non-NPF is a major finding in our manuscript and should be in the abstract. We also think that the difference in the back trajectories is a relevant difference; therefore, we want to keep the statement in the abstract. We changed the last sentence of the abstract in the revised manuscript, line 65-68: The two major differences between NPF days and non event days are two. A factor of 2 lower condensation sink on NPF days and different air mass origins for the NPF days compared to non event days. We followed the suggestion by Referee 1 and included a new Figure (Fig. 10) in the revised manuscript, which shows the calculated back trajectories on a map. The map shows actually that the trajectories on do not pass over Manaus on NPF days. The sentence 'Nevertheless, all air masses pass over Manaus before reaching the measurement site.' has been deleted from the revised manuscript. The first set of back trajectory calculations was made without looking at the map, which lead to the wrong conclusion.

Referee comment 6) Lines 221-222 – You say "The vertical mixing can be enhanced during the wet season due to convective clouds." Are you saying that convective clouds lift the mixed layer or that convective clouds lift particles out of the mixed layer or something else? Clouds formed at the top of a mixed layer will tend to cool below, which does not help the development of a mixed layer.

Reply: We agree with the referee that this sentence slightly confusing. We were trying to say that convective clouds lift particles out of the mixed layer. We re-phrased the sentence in the revised manuscript, line 289-290: 'The vertical mixing can be enhanced during the wet season as particles are lifted out of the mixed layer due to convective clouds.'

Referee comment 7) Lines 275-279 – This may be true for inside the canopy, but not for

outside the canopy. Please clarify. Also, why would the pattern outside of the canopy not reflect biomass burning and wet deposition more than that inside the canopy? Reply: We do believe that the statement is true for both outside and inside the canopy. The chapter title in this paragraph. 'Number concentrations of ions and particles at the two sites' includes both measurement sites. It seems like this statement is not fully clear, so we rephrased the sentence in the revised manuscript as follows, line 361-365: 'Additionally, the wet and dry seasonality characteristic for the Amazon (Rissler et al. 2006, Martin et al. 2010a) can be observed in the concentration of the large ions (4-20nm): the local biomass burning during the dry season seems to increase large ion concentrations, whereas during the wet season their concentrations are decreased most likely due to wet deposition and reduced source strengths.' Referee comment 8) Lines 281-282 - That appears to be true for the wet season, but the factor is less than 2 during the dry season. Did you mean "up to a factor of 3"? Reply: we agree with the referee that the statement should be 'The average concentrations of 4 – 20 nm particles were up to a factor of 3 higher in comparison to the less polluted site (T0t).'

Referee comment 9) Lines 287-288 – The 4-20 nm ions are not shown in the Figure 2 I have. Reply: We apologize for the mistake in the manuscript. The Figure has been changed during the writing process of the manuscript. The sentence has been re-phrased as follows in the revised manuscript, line 367-368: 'Figure 2 shows the monthly variability of particles in two size ranges (0.8-2nm, 2-4 nm) for the 2011-2014 period. The cluster ions had a median concentration of 814 cm-3 and 968 cm-3 (wet) and 605 cm-3 and 765cm-3 (dry) for negative and positive ions, respectively.'

Referee comment 10) Line 301 – "Oct-Dec for both seasons"? Oct-Dec is a season (fall). Specify wet and dry seasons. Reply: we agree with the referee that this sentence is confusing. We removed the sentence completely from the revised manuscript. We agree with the referee that the seasons should be specified in our manuscript. We added a sentence in the Methods section, line 139-142: 'Wet and dry season in the Amazon are Dec-March and June-September respectively (Martin et al, 2010a). Due

to the measurement periods available for our dataset, we define the dry season as dry and transition season Apr-Oct.'

Referee comment 11) Line 305 – On line 214 the dry and transition season is April to September, whereas here it is Apr-Oct. Please correct. Reply: this sentence was removed in the revised manuscript. The whole paragraph was changed following the suggestions of Referee 1. Line 401-405: 'These values are comparable, for example, to intermediate and large ion concentrations found in coastal Mace Head (Vana et al. 2008) outside the periods of rain or active NPF. In general, the positive cluster ion concentrations are higher in all the cluster ion and intermediate ion size classes for all the months. Table 2 summarizes the annual concentrations of ions and total particles for the three size bins.'

Referee comment: 12) Line 311-312 – Cluster ions are not shown in Figure 3. Where are we supposed to view this? Reply: this has been removed in the revised manuscript, as we do not present the Figure in the final manuscript. The paragraph in the revised manuscript was re-phrased as follows, line 406-409: 'Differences between the wet (Dec-Mar) and dry and transition season (Apr-Oct) were also observed in the diel cycle of the ion and particle concentration. Positive and negative cluster ion concentrations were, on average, higher during the wet season compared to the dry season as shown in Table 1.'

Referee comment 13) Lines 321-323 – Again, 4-20 nm ions are not shown in Figure 2 Reply: this has changed in the revised manuscript. The paragraph was re-phrased as stated in the reply to the previous Referee comment 12.

Referee comment 14) Line 358 – What do you mean when you say that "negative ions are smaller than positive ions"? Do you mean fewer in number? Reply: We mean that the negative ions are smaller in size compared to positive ions. The sentence has been re-phrased in the revised manuscript as follows, line 478-484: 'Rain-induced bursts are likely a result of a balloelectric effect, in which splashing water produces intermediate

ions such that the negative ions are smaller in size than the positive ions (Horrak et al., 2005, Hirsikko et al., 2007, Tammet et al., 2009). The duration of the 579 rain events varied from a couple of minutes to 22 hours, with over half the rain events lasting for two hours or less. The rain events were more common during the wet season, peaking in August which can be considered as transition season (Fig. 5) when also the median rain intensity was higher.'

Referee comment: 15) Lines 374-376 and figure 6 – For the ions in the 0.8-2 nm particles, it looks like they simply turn on at rain intensities above 1. Reply: We made Figure 6 in order to show the relation between rain intensity and ion concentrations. At rain intensities below 1 mm/h the ion concentration especially in the cluster ion size range only contains the natural in background as they are produced via radon decay or galactic cosmic rays. The background cluster ion band can be observed worldwide, yet the concentrations depend on the location as it depends on the sources and sinks for the ions.

Referee comment 16) Figure 7 – Indicate which axis corresponds with which particle size class in Panel B; presumably, the LH axis is 6-10 nm. Reply: we agree with the referee that the Figure is confusing. The Figure was changed to improve the clarity in the revised manuscript.

Referee comment 17) Line 379 – "followed by a second one at about 11:00". Here, indicate the relative difference in rain intensity. Reply: The second rain intensity peak was lower than the first one in this example. The first one was about 40 mm/h and the second one about 10 mm/h. So, the difference was about 30 mm/h. We added the relative difference in the revised manuscript, line 502-503. 'Rain events were evident also when looking at the total particle concentrations measured by the NAIS, as depicted in Figure 7. The first rain event showed a maximum of about 40 mm h-1 and the second one about 10 mm h-1.'

Referee comment 18) Figure 7 and lines 385-395 – This is a very interesting set of

observations. If particles descending with the rain were responsible for the increase in 6-10 nm particles above the canopy, how do you explain the apparent evolution of 6-10 nm particles to 10-20 nm over a few hours? Given the roughly 3 orders of magnitude difference in particle number concentrations from ground to above canopy and the potential canopy filtering you mention, why instead is it not possible that the few 6-20 nm particles above the canopy were due to the rain-induced particles mixing and filtering upwards? Reply: There are two main points about Figure 7. The first one should show a clear correlation between the rain intensity and increase in ion concentrations inside the canopy. This effect is due to the splashing of the water droplets on the leaves mainly of the trees. The water droplets explode and release high amounts of small ions. This phenomenon has been observed and explained by Tammet et al, 2009. The second effect is that these ions seemingly do not contribute to the total particle population as the DMPS which is measuring from above the canopy does not show an according increase in particle number concentration. We assume that this effect is due to the filtering effect by the rainforest canopy. We cannot say very much about the source of the particle that are seen by the DMPS, but from our current knowledge it is likely that they are transported via downdraft from production at convective cloud outflow regions (Wang et al, 2016). We rephrased the paragraph in the revised manuscript, line 509-518: The 10-20 nm particle concentration showed first a decrease followed by a slight increase up to $\sim$35 cm-3, peaking later than the 6-10 nm particles. However, it is unlikely that these 10- 20 nm particles originate from the same rain-induced burst as seen inside the canopy, as there is no apparent particle growth from the NAIS measurements. It is unlikely that those particles survive until the top of the canopy, as the tree leaves would filter them out. Wang et al. (2016) reported that nucleation mode particles produced in cloud outflows will be transported down with the rain, such that they can be observed at the ground level as an increase in nucleation and Aitken mode concentrations (Dp <50 nm). The appearance of 6-10 nm particles with its peak concentration, could present a similar scenario of small particles brought down from the free troposphere.'

Referee comment 19) Table 3 and lines 404-406 - Table 3 shows 65 and 49 for a total of 114, while you state 64 and 46 and 113. Please correct. Reply: the numbers have been corrected in the table.

Referee comment 20) Figure 9 – On either side, you show four panels. The top two are labelled ions and the bottom two are labelled total particles, which is consistent with the text. In the caption, we are led to believe that the top three are ions. Please correct. Reply: we agree the Figure and description was unclear. The Figure and caption were updated in the revised manuscript.

Referee comment 21) The RH side of Table 5 is cut off in my copy. Reply: we are sorry about that. Table 4 is better readable in the revised manuscript.

Referee comment 22) Line 520 – Should be Jan-March for wet season? Reply: We agree with the referee. The statement has been corrected in the revised manuscript, line 691-692. 'We observed eight NPF events showing particle growth at site T3 outside the canopy during Jan-March 2014, which is during the wet season.'

Referee comment 23) A couple of more general comments: Is there some sort of summary connecting the ion concentrations with NPF that can be drawn? The rain-induced events are prominent, but we are not given any sense of how important these might be. For example, is there any evidence that a significant number of rain-induced particles survive to become CCN size, or is Figure 7 the best example of their potential longevity? Reply: Our analysis has clearly shown that the connection between rain and NPF events is not clear. There was no rain observed during any of the NPF events, yet sometimes there was rain in the evening after or shortly before the start of an NPF event. That indicates that the rain clears the air of pre-existing particles and therefore the conditions for NPF events to happen are favorable. To our current understanding, the increase in ion concentrations due to rain events that were mainly observed inside the canopy do not significantly contribute to the production of bigger neutral particles as there is no concomitant increase in neutral particle concentrations as measured by

the DMPS, which is sampling from above the canopy (see Figure 7). We believe that the ion production due to rain is mainly an inside canopy effect and the ions are filtered out by the canopy and therefore do not survive until they would be able to reach bigger sizes. We did observe some rain events that lasted up to 20 hours but still we did not observe any increase in neutral particle concentrations above the canopy.

References de Sá, S. S., Palm, B. B., Campuzano-Jost, P., Day, D. A., Newburn, M. K., Hu, W., Isaacman-VanWertz, G., Yee, L. D., Thalman, R., Brito, J., Carbone, S., Artaxo, P., Goldstein, A. H., Manzi, A. O., Souza, R. A. F., Mei, F., Shilling, J. E., Springston, S. R., Wang, J., Surratt, J. D., Alexander, M. L., Jimenez, J. L., and Martin, S. T.: Influence of urban pollution on the production of organic particulate matter from isoprene epoxydiols in central Amazonia, Atmos. Chem. Phys., 17, 6611-6629, https://doi.org/10.5194/acp-17-6611-2017, 2017

Hirsikko, A., Bergman, T., Laakso, L., Dal Maso, M., Riipinen, I., Horrak, U. and Kulmala, M., Identification and classification of the formation of intermediate ions measured in boreal forest, Atmos. Chem. Phys., 7, 201-210, 2007.

Hõrrak, U., Tammet, H., Aalto, P. P., Vana, M., Hirsikko, A., Laakso, L., and Kulmala, M.: Formation of charged particles associated with rainfall: atmospheric measurements and lab experiments, Rep. Ser. Aerosol Sci., 80, 180–185, 2006.

Martin, S. T., Andreae, M. O., Althausen, D., Artaxo, P., Baars, H., Borrmann, S., Chen, Q., Farmer, D. K., Guenther, A., Gunther, S. S., Jimenez, J. L., Karl, T., Longo, K., Manzi, A., Müller, T., Pauliquevis, T., Petters, M. D., Prenni, A. J., Pöschl, U., Rizzo, L. V., Schneider, J., Smith, J. N., Swietlicki, E., Tota, J., Wang, J., Wiedensohler, A., and Zorn, S. R.: An overview of the Amazonian Aerosol Characterization Experiment 2008 (AMAZE-08), Atmos. Chem. Phys., 10, 11415–11438, doi:10.5194/acp-10-11415-2010, 2010b. Tammet, H., Hõrrak, U., and Kulmala, M.: Negatively charged nanoparticles produced by splashing of water, Atmos. Chem. Phys., 9, 357-367, doi:10.5194/acp-9-357-2009, 2009. Thalman, R., de Sá, S. S., Palm, B. B., Barbosa,

H. M. J., Pöhlker, M. L., Alexander, M. L., Brito, J., Carbone, S., Castillo, P., Day, D. A., Kuang, C., Manzi, A., Ng, N. L., Sedlacek III, A. J., Souza, R., Springston, S., Watson, T., Pöhlker, C., Pöschl, U., Andreae, M. O., Artaxo, P., Jimenez, J. L., Martin, S. T., and Wang, J.: CCN activity and organic hygroscopicity of aerosols downwind of an urban region in central Amazonia: seasonal and diel variations and impact of anthropogenic emissions, Atmos. Chem. Phys., 17, 11779-11801, https://doi.org/10.5194/acp-17-11779-2017, 2017.

Wang J., Krejci R., Giangrande S., Kuang, C., Barbosa H. M. J., Brito J., Carbone S., Chi X., ComstockJ., Ditas F., Lavric J., Manninen H. E., Mei F., Moran-Zuloaga D., Pöhlker C., Pöhlker M. L., Saturno J., Schmid B., Souza R. A. F., Springston S. R., Tomlinson J. M., Toto T., Walter D., Wimmer D., Smith J. N., Kulmala M., Machado L.A. T., Artaxo P., Andreae M. O., Petäjä T. & Martin S. T., Amazon boundary layer aerosol concentration sustained by vertical transport during rainfall, Nature, doi:10.1038/nature19819, 2016.

Please also note the supplement to this comment:
https://www.atmos-chem-phys-discuss.net/acp-2017-782/acp-2017-782-AC2-supplement.pdf

[Figure]

**Fig. 1.** Figure 10: median back trajectories for NPF (blue) and non NPF (red) days. The trajectories were calculated 24hours backwards arriving at 09:00 local time at 500m a.s.l. at the measurement site.

[Figure]

**Fig. 2.** Figure 7. Example for a rain-induced event for total particles (DMPS). The DMPS measurements are taken above the canopy (60 m height), NAIS measurements are inside the canopy. Panel (a) shows the DMPS

[Figure]

**Fig. 3.** Figure 9. Diel cycle of ions measured outside the canopy by the NAIS (small: 0.8–2 nm; intermediate: 2–4 nm; The lowest two panels show the total particles (large: 4–20 nm) from the NAIS and total par

---

## Referee Report (RR1)

I thank the authors for addressing my concerns. Nevertheless, there are still open questions, incorrect statements, internal inconsistencies and technical mistakes.
The high amount of mistakes in form and content as well as the incorrect citations leave me with serious concerns about the data quality, the analysis and manuscript preparation.
The updated manuscript is not improved significantly and does not meet the standards of ACP.

A thorough and careful major revision of the data analysis and the entire manuscript is necessary before considering publication.

I divide my comments into two major parts. First, I address the author comments based on their supplementary material. Afterwards, I address specific comments in the updated manuscript.
I put all author comments by Wimmer et al. in blue italic font.

**Replies to Author comments:**

*Replies to referee #1*

*We thank the referee for the careful revision of the manuscript 'Direct observations of molecular clusters and nucleation mode particles in the Amazon'.*
*The comments improve the current manuscript. We will address all the comments and concerns in detail as shown below/ as in the following paragraphs.*

*General comments.*
*We thank the reviewer for suggesting the comparison of the two research sites. This issue has been addressed carefully in the revised manuscript.*

*The identity of specific sentences in the current manuscript were a mistake. We have re-phrased the identical sentences from previous publications in the revised manuscript.*
*We address the specific comments of the referee here below.*

*Referee comment:*
*There are specific sentences and complete text passages which are identical to Martin et al., 2016. The following list is not necessary complete. The authors should make sure that further text passages similar to other work are referenced correctly. I encourage to use the similarity report provided by the iThenticate plagiarism screening service.*

*Reply: All the identical text passages to previous publications have been re-phrased in the revised manuscript.*

**Reply to author comment:**
Rephrasing the original sentences does not solve the citation issues. If you rephrase a sentence from a different source, you still have to cite the original source. I am concerned the authors do not take reasonable care to check their citations.
Make sure all the identical sentences and text passages mentioned in my first comments are now correctly referenced - even if rephrased.

*Specific comments:*
*Page 4, lines 134:*
*The authors state that T3 is located in a pristine environment. According to e.g., Martin et al., 2016 T3 (time points three) is located downwind of the pollution in a pasture area. I suggest to not use 'pristine' in this context.*
*Reply: We agree with the referee. The term pristine has been removed from the revised manuscript*

**Reply to author comment:**
You mention the term pristine already in the abstract. Your answer 'The term pristine has been removed from the revised manuscript' is incorrect and misleading. Be precise what parts of the manuscript are changed.
As referee 2 already pointed out, you have to clearly define 'pristine'.

*Page 4, lines 118:*
*"T0t is mostly unaffected by the Manaus pollution and is surrounded by dense rainforest. It allows the characterization of an almost completely undisturbed natural environment"*
*- Did the authors filter for pollution affected periods? If so, what are the filter criteria?*

*Reply: In the general data analysis, we did not filter for pollution affected periods, since we report average values for the whole measurement period and wet/dry season specifically. However, for the analysis of the NPF events, pollution events would appear in the NAIS/SMPS data as elevated aerosol concentrations in the accumulation mode. Also, the calculation of the condensation sink gives a good criterion for polluted days, which is clearly higher on non NPF days.*
*Since we observed two nucleation events, with GR of approximately 10-20 nmh -1  and about 1 nm h-1, it might be that the days with the higher GR are days which are more influenced by the Manaus pollution plume. Since the sulfuric acid concentrations seemed to be about the same on days with high and low GR, we may assume that the Manaus pollution is not the main factor influencing the air masses.*

**Reply to author comment:**
The authors argue that '.. pollution events would appear in the NAIS/SMPS data as elevated aerosol concentrations in the accumulation mode.'.

According to Kuhn et al., 2010, the Manaus pollution plume consists to a significant degree of fuel combustion. Kuhn et al., 2010 found CN concentrations up to several 10000 particles per cubic centimeter - the majority of these particles were likely smaller than 40 nm. Hence, pollution events can influence NAIS and SMPS data to highly variable degrees in a broad size range not only in the accumulation mode. The author's argumentation that filter criteria are not necessary is not convincing.

Furthermore, in your abstract you insist to present measurements under pristine conditions. You further state, that even the parallel-wind station T0t site is affected by the Manaus pollution plume about once per week. To my understanding, pristine refers to undisturbed, clean or natural conditions.
Without a proper filter to exclude pollution sources, your results can not be considered as pristine. Again, please provide a clear definition for 'pristine'.
Presenting average values does not help in this case since these average values will be affected by pollution episodes to variable degrees.

*Page 4, lines 124:*
*The introduced DMPS measurements are performed using an inlet line above canopy. Nevertheless, the section is called 'inside canopy measurements' which is confusing. I further wonder if there are any comparisons of the DMPS and NAIS during the 3-year period to confirm the quality of measurements.*

*Reply: The section has been re-named to 'Measurements inside the rainforest' to avoid confusion.*
*We changed the classification of the two sites in the whole manuscript accordingly. T0t is called inside rainforest site and T3 pasture or outside rainforest site.*
*The instrumentation was calibrated before shipping to the campaign and regular maintenance including flow adjustments and zero checks were performed.*

*Page 5, lines 136:*
*"The site is located in a clearing of the rainforest." According to Martin et al., 2016 the site is located in a pasture area (2.5 x 2 km) outside the rainforest. I suggest to rephrase the text accordingly from 'outside canopy' to 'outside forest' or 'pasture site'.*

*Reply: We thank the referee for the suggestion. The text has been rephrased accordingly, line 177-180: 'The site is an open pasture site, where the Manaus pollution plume regularly intersects and the rainforest canopy did not hinder mixing. Due to the site location, T3 is either a pristine environment or highly influenced by the Manaus pollution plume, mainly depending on the wind direction.'*

**Reply to author comment:**
I thank the authors for clarification.

*Page 6, lines 180:*
*A description of the applied inlet system for the PSM would be interesting for future studies under high rh conditions.*
*Reply: We agree with the referee. A description has been added to the revised manuscript, line 236-243: The inlet system consists of a core sampling probe combined with a sintered tube. The core sampling probe consists of two cylindrical tubes with different outer diameters (10 mm and 6 mm). The larger diameter of the outer tube allows up to 10 Lpm total laminar flowrate, to minimize diffusional losses. The inner tube is directly attached to the PSM with an airflow of 2.5 Lpm. The excess airflow is discarded into an exhaust line (Kangasluoma et al, 2016). Downstream of the core sampling line is a sintered tube where dry pressurized air is introduced. The water molecules in the sample flow are pushed towards the outer walls of the sinter material by diffusion, drying the airflow.*

**Reply to author comment:**
I thank the authors for adding these information.

*Page 6, lines 183:*
*"Laboratory studies have shown that the RH affects the counting efficiency of the PSM drastically" - Please provide references.*
*Reply: the sentence has been rephrased as follows:*
*Line 244-246: Laboratory studies have shown that the RH affects the counting efficiency of the PSM drastically (higher sensitivity at smaller sizes at higher RH; Kangasluoma et al. 2013, Iida et al, 2009).*

*Page 7, lines 203:*
*"The DMPS data reported here is qualitative but not quantitative." - Please specify if*
*there were problems with this instrument. Quantitative SMPS data are discussed in*
*e.g., section 3.2.*

*Reply: the issue is addressed more precisely in the revised manuscript. Since the particle*
*losses in the sampling line due to diffusion are not precisely known, the SMPS data has*
*not been corrected for those losses. Hence, for the data shown in Figure 7, where the*
*concentrations of 6-10 nm and 10-20 nm are shown, we feel comfortable only at making*
*assumptions based on the trend of the data but not absolute numbers.*
*We added the following sentence to the revised manuscript:*
*Page 7, lines 264-267: 'The DMPS data reported here are qualitative, not quantitative, as*
*the losses due to diffusion in the sampling line are not precisely known and therefore not*
*taken into account in the data presented later in this manuscript.'*

**Reply to author comment:**
Even a rough estimate would already help to put your integral DMPS data into context.
Are these concentrations underestimated by a factor of 2 or an order of magnitude?

*Page 7, lines 220:*
*The planetary boundary layer development is probably different for pasture and rainforest*
*sites. Can you please comment on that?*
*Reply: The local features and land-use affect the development of convective boundary*
*layer as well as their emission spectra in terms of volatile organic compounds are*
*different. In the morning, the boundary layer develops more rapidly in the pasture area due*
*to lower evapotranspiration and the sensitive heat flux is dominating. This induces a more*
*rapidly growing mixed layer, causing more efficient vertical mixing of precursors and*
*aerosols. Also, photochemistry is more pronounced in (semi) open area than under the*
*canopy. However, during the daytime the small-scale variability in boundary layer*
*dynamics and in VOC concentrations tends to even out. The rapid oxidation chemistry*
*remains characteristic for each site.*
*We added a sentence in the revised manuscript. Page 8, line 293-297: 'The*
*boundary layer development is also different at the two different measurement sites. It*
*develops more rapidly in the pasture area, causing a more efficient vertical mixing*
*compared to the site enclosed by rainforest. From our observations, we conclude that*
*the main differences in the dynamics of the aerosol particle population at the two*
*measurement sites is due to the 'umbrella effect' of the rainforest canopy.*

**Reply to author comment:**
I do not understand what the authors are trying to say with: 'However, during the daytime
the small-scale variability in boundary layer dynamics and in VOC concentrations tends to
even out.' Can you please provide references?

*Page 8, lines 234:*
*"We observed an unexplained increase in the concentrations of the cluster ions in the*
*NAIS towards the end of October 2013 to January" - Can you please comment on*
*possible reasons for that drift? Is it possible that this drift continued after moving to T3?*

*Reply: we carefully looked at the flow rates and other NAIS technical data that could give*
*some input, but we could not find any clear indicator of an instrumental drift. The drift*
*continued after moving to T3, which is why we corrected all the data after we observed the*
*drift for the first time accordingly. We attribute the drift is caused by a slow change in the*

*differential mobility analyzer flow rates and charger ion filtration that cause erroneously some of the corona charger generated ions to penetrate into the detectors.*

*We explain this with the following sentences on page 8, lines 306-323 in the revised manuscript:*
*'We observed an unexplained increase in the concentrations of the cluster ions in the NAIS towards the end of October 2013 to January 2014 at the T0t site. This increased level continued when the NAIS was taken to the T3 site. We consider this drift instrumental. By comparing the 2014 concentrations of the NAIS channels to those prior to the increase (January 2012 and 2013), a correction factor of 1.8 was applied to the 4 smallest size channels of the NAIS (0.8-1.25 nm) to account for the drift for the subsequent data.'*

**Reply to author comment:**
I have a few questions on this. Does the drift affect positive and negative cluster ions? Furthermore, you relate the drift in cluster ions to ions generated by the corona charger. According to Manninen et al., 2016, all parts of the preconditioning unit are switched off while naturally charged ions are measured. How can then natural ions be affected?

*Page 9, lines 276:*
*"the biomass burning during the dry season is expected to increase large ion concentrations" - Please provide a reference*

*Reply: we rephrased the sentence as follows:*
*Page 9, lines 361-365:*
*'Additionally, the wet and dry seasonality characteristic for the Amazon (Rissler et al. 2006, Martin et al. 2010a) can be observed in the concentration of the large ions (4-20nm): the biomass burning during the dry season seems to increase large ion concentrations, whereas during the wet season their concentrations decreased, most likely due to wet deposition and reduced source strengths.*

*Page 10, lines 287:*
*"Figure 2 shows the seasonal variability of ions and particles in the three size ranges (0.8-2nm, 2-4 nm and 4-20 nm)" - the lowermost panel in Fig. 2 is missing.*
*Reply: This is a mistake. During the writing process, we decided not to show the 4-20 nm size range as it does not add any additional valuable information. The sentence was changed in the revised manuscript as follows:*
*Page 10, lines 373-374:*
*'Figure 2 shows the monthly variability of ions and particles in two size ranges (0.8-2nm, 2-4 nm) for the 2011-2014 period.'*

*Page 10, lines 305:*
*In this paragraph it is not clear to which figure or table the authors refer to. Some examples:*
*"Positive and negative cluster ion concentrations were, on average, higher during the wet season compared to the dry season."*
*"Additionally, cluster ions (0.8-2 nm) showed slightly higher concentrations in the morning and evening, compared to other times of the day"*
*"A dip in the median ion concentration after midday coincides with a higher median concentration of large ions, which is a sign of a larger sink for cluster ions."\newline*
*"Lastly, 4-20 nm ions peaked at around midday during the wet season, while their diel pattern was more irregular during the dry season."*

*Reply: this paragraph has been deleted from the revised manuscript. The numbers refer to a Figure that has been removed from the final manuscript, as we decided to only show the particle concentrations, as the data shows a very similar behavior as the ion data. The ion data does not add any additional information to the manuscript.*

*Page 11, lines 343:*
*"The median total particle concentrations were about a factor of two higher during dry season (about 1500 cm-3) compared with the wet season (about 700 cm-3)." - In table 1 different values are shown. Furthermore, large particle (4-20 nm) concentrations are very similar to CPC measurements (> 10 nm), implying that on average all particles are in the size range between 10 and 20 nm.*
*Also, the average particle concentrations (4-20 nm) at T0t (250-800, for the wet season) compares well to total particle concentrations (e.g., in 10-500 nm size range) reported in earlier studies (e.g., Martin et al., 2010a, Martin et al., 2010b, Zhou et al., 2002). This again implies that the size distribution is dominated by nucleation mode particles, which is in contrast to the same mentioned references.*

*Reply: The numbers reported in the text are a mistake. The sentence has been re-phrased as follows (page 11, lines 426-428 in the revised manuscript):*
*The median total particle concentrations were about a factor of 1.5 higher during wet season (about 1000 cm -3 ) compared with the dry season (about 700 cm -3 ).*
*The presented manuscript is (to our knowledge) the first comprehensive study of small ions and particles in the Amazon basin. We agree with the referee that from looking at those numbers, we could conclude that the aerosol particle population in the Amazon is dominated by the nucleation mode.*
*Nevertheless, we should be careful since previous studies have not been focusing on nucleation mode particles. All the numbers presented in the current manuscript for the T0t site are directly from the measurements with the NAIS. Whereas the previous results have been using different instrumentation and the measurement locations have been different. We think that from our current knowledge we cannot conclude that the aerosol particle population in the Amazon is dominated by the nucleation mode.*

**Reply to author comment:**
The authors agree, that from their presented particle number concentrations from NAIS and CPC measurements one could conclude that their findings indicate a dominating nucleation mode in the Amazon aerosol particle number size distribution but at the same time they do not. This is confusing.

I would like to outline my concerns based on the results shown in Table 1:

1. At T3, the measured particle number concentrations from NAIS (4-20 nm) and CPC (> 10 nm) are on average very similar. If both instruments are comparable this means, almost all or at least a very large amount of the measured particles must be in the size range 10 - 20 nm. Hence, it follows from these measurements, that the aerosol population is dominated by nucleation mode particles. If true, this has to be supported by further size resolved measurements, since it is a stark contrast to the mentioned existing literature (see my first review, e.g., Martin et al., 2010a, Martin et al., 2010b, Zhou et al., 2002).

If one cannot draw this conclusion, NAIS and CPC measurements are not consistent. In any case, a detailed paragraph with a thorough discussion on this discrepancy (if it is one) has to be added to this manuscript.

2. At T0t this comparison (CPC, NAIS) is not possible. Nevertheless, comparing the particle number concentrations shown in this manuscript (NAIS, 4-20 nm) with total particle number concentrations in the mentioned references, one can again conclude, that the majority of particles is in the nucleation mode size range. The authors argue, that the mentioned references did not focus on new particle formation. But clearly, these authors used instrumentation sensitive to the nucleation mode size range.

The authors have to put their findings (at least for the particle concentration data) into context of existing results and have to discuss their high concentration of nucleation mode size particles.

*Page 12, lines 361:*
*"The rain events were more common during the wet season (Fig. 5) when also the median rain intensity was higher." According to Fig 5, the median rain intensity is highest in August.*
*Reply: The sentence has been rephrased as follows: p. 12, lines 482-483 in the revised manuscript:*
*'The rain events were more common during the wet season, peaking in August which can be considered as transition season (Fig. 5; Martin et al, 2010) when also the median rain intensity was higher.'*

**Reply to author comment:**
The rephrased sentence is confusing. How can the rain events be more common in the wet season, when they 'peak' in the transitional season? Also, the reference is unclear. Do you reference Fig. 5 in Martin et al., 2010?

*Page 11, lines 377 and following:*
*In section 3.2 the authors describe a very interesting and scientifically significant phenomenon of increased particle and ion concentrations during rain. Concentrations increase by 2 orders of magnitude towards more than 10000 particles/ions per cubic centimeter. In the following discussion, the authors mention that the particle concentration (nucleation mode size) above canopy (SMPS) does not increase accordingly. Instead, particle concentration increases only by 20 particles per cubic centimeter (6-20 nm size range), strongly contrasting the conditions below. They conclude that the high particle/ion concentration is a below canopy phenomenon. Furthermore, these nucleation mode particles are not able to leave the canopy which is acting as an umbrella preventing mixing.*
*In contrast, the presented diurnal variation suggests that mixing and planetary boundary layer development is efficient (although less efficient as compared to the pasture site). Also, the authors argue that they are able to measure ions and particles related to transported biomass burning plumes (page 9, lines 275). Why are those particles able to be mixed into the canopy. It is hard to believe that the forest canopy can maintain such a strong gradient of particle number concentration.*
*Please justify your statement.\\*

*Reply: Earlier studies have shown that rain and particularly shattering of water droplets will result in high concentration of ions (e.g. Tammet et al., 2009). Typically, these effects are not seen with aerosol instruments as the ions are neutralized in the measurement process. Our main point here is that this increase in ion concentrations is mainly an effect that can be observed inside the canopy as the ions that we observe are produced by splashing of the water droplets on the tree leaves. Those ions will not survive until the*

*measurements by the DMPS as it is sampling from above the canopy and they ions are filtered out by the leaves before reaching the inlet of the DMPS. From the current measurements, we cannot make any statement of the source of the larger neutral particles that are seen by the DMPS above the canopy. It is likely that they are produced in cloud outflow regions and due to strong downdrafts entrained back into the mixing layer (Wang et al, 2016). Most likely the increase of 4-20nm ions during the dry season is a combination of local biomass burning sources and a decrease in wet deposition.*

*The sentence has been re-phrased in the revised manuscript:*
*Page 9, lines 361-365: 'Additionally, the wet and dry seasonality characteristic for the Amazon (Rissler et al. 2006, Martin et al. 2010a) can be observed in the concentration of the large ions (4-20nm): the biomass burning during the dry season seems to increase large ion concentrations, whereas during the wet season their concentrations decreased, most likely due to wet deposition and reduced source strengths.*

**Reply to author comment:**
The rephrased sentence does not address my main criticism. My main concern is that the 2 orders of magnitude increase for the particle number concentration is not visible at all above the canopy.
I agree that the increase in ion concentration will not be detected by the DMPS, but according to you methods section, the DMPS should be able to detect particles larger than 6 nm (taking into account the CPC cutoff and the inlet losses as stated by you).

The forest canopy is certainly hindering mixing. Nevertheless, it is likely that a certain amount of these small particles (if generated by droplet splashing) is already produced at the top of the canopy. At least a fraction of these small particles does not have to go through the canopy and should therefore appear in the DMPS measurements.

I put up the comparison with the ions produced by biomass burning for another reason. You argue that those ions are able to pass the canopy, but the others are not - why?

*Page 15, lines 45:*
*Please consider to show the results of your backward trajectory analysis in a map.*

*Reply: We thank the referee for the suggestion, as it improves the manuscript. We have added a Figure showing the map to clarify the back trajectory calculations.*
*We also rephrased the sentence as follows, line 594-598: 'These air masses all originate from upstream of the Amazon river, where the NPF day air mass originate from further north, which is an area with dense rainforest. The results of the back trajectory calculations are shown in Figure 10. The red line shows the median of an ensemble or the non event days and the blue line for NPF days.*

**Reply to author comment:**
I thank the authors for adding these information.

*Page 15, lines 459:*
*In Fig. 10 a new particle formation event is shown: Please consider to add SMPS contour plots and SMPS particle number concentrations in the nucleation mode size range. Statistical information of SMPS nucleation mode particle number concentration will add further valuable information to Figure 9 and Tables 1 and 4.*
*The absence of the forest canopy at T3 gives the opportunity to combine NAIS and SMPS measurements, which allows to investigate the entire evolution of the submicron*

*aerosol population.*
*Reply: we thank the referee for this suggestion. The SMPS Figure has been added to the Figure.*

**Reply to author comment:**
I thank the authors for adding these information. Nevertheless, the SMPS plot shows a linear Dp - axis which makes it hard to identify structures below 100 nm.

*Page 17, lines 510:*
*"Similar, but weaker, rain-events were found at the site outside the rainforest canopy (T3)." - weaker in terms of what?*
*Reply: We have re-phrased the sentence as follows in the revised manuscript:*
*Page 17, lines 679-683: 'Similar rain-events were found at the pasture site (T3). The production of small (0.8-2 nm) and intermediate ions (2-4 nm) during rain events reached a maximum of 10 4 cm -3 at the pasture site, where it was one order of magnitude higher at the T0t site. Large ion concentrations reached similar concentrations during rain events at both measurement sites.'*

**Reply to author comment:**
Do you mean 'while' instead of 'where'?

*Technical comments related to Figures*
*The boxes refer to the 25 th -75 th percentile.*
*Reply: The whiskers show the extreme values of the data set which are not considered outliers.*

**Reply to author comment:**
This answer is very imprecise and raises serious concerns about the quality of the data analysis. There must be a clear (mathematical) definition for the shown whiskers. Furthermore, all figure captions still state, that the whiskers are related to the 25th and 75th percentile. I doubt that this is correct.

*The tables and Figures have been changed according to the suggestions of the referee in the revised manuscript.*

**Reply to author comment:**
This is not correct - the boxplots are not described correctly.

*Fig 4:*
*number concentration of small positive and negative ions disagrees by a factor of 2. According to Manninen et al., 2016 there should be an agreement within 20%. Please comment on the instrument performance and data quality.*

*Reply: Table 1 shows a very good agreement between the positive and negative ion concentrations. We believe that the difference seen in Figure 4 is due to a problem with the instrument performance, which might be different on certain days, but which does not affect the overall good instrument performance and data quality.*

**Reply to author comment:**
Table 1 shows averages. To proof that two variables agree within a certain range it might be necessary that their averages agree but it is not sufficient.
Furthermore, for T0t (dry season) positive ion concentration is on average 26% larger than negative ion concentration. The answer is not convincing.

**Additional specific comments related to the updated manuscript:**

Page 1, l. 38:
'In this work, pristine refers to CCN concentrations of a few hundred cm -3 .'
The authors do not present CCN measurements. This statement is out of context and does not help to classify the presented results as pristine or not pristine.

Page 2, l. 47:
'T0t is influenced by pollution about once per week, where T3 on the other hand is reached once per day/once per every second day, especially in the afternoon (Martin et al., 2010b supplementary material, Thalmann et al, 2017, de Sa et al, 2017).'
This sentence is not clear to me. Does it mean, pollution arrives mainly in the afternoon, or does it mainly affect only the afternoon?

Page 3, l. 104:
'The different meteorological and aerosol dynamical conditions during the wet and the dry season in the Amazon basin, offer an interesting natural environment for studying aerosol particle dynamics.'
Different aerosol dynamical conditions make it interesting to study aerosol dynamics - this sentence is a tautology.

Page 4, l. 148:
'Manaus is the capital of the state of Amazonia, Brazil and is located where the Rio Negro merges with the Solimoes river which then form the Amazon river. The city with more than 2 million inhabitants is the seventh biggest city in Brazil and is surrounded by 1500 km of forests in all directions (IBGE, 2015; Martin et al., 2016)'
This is partly a repetition of the introduction.

The methods section misses a detailed description of the inlet design, height and aerosol treatment (e.g., drying) at T3 and T0t.

Page 10, l. 368:
'The environmental variables were relatively similar between the two sites, the temperature and RH being slightly lower at the pasture site compared with the inside rainforest site.'
Isn't this counter-intuitive. Why is the temperature lower outside the forest. According to your table this is also not correct.

Fig. 3:
The diurnal cycles show 25 hours. There is something wrong with the data analysis.

Fig. 3:
The lower whiskers end below zero. This means that a large amount of your measurements is close to or below zero particles per cubic centimeter. Is this correct? Is it an artifact? You are stating, that you clean your data.

Fig. 5:
The figure shows number of days with and without rain. Obviously, the number of observations per month varies. That makes it hard to compare these numbers. Information about the total number of days taken into account are missing.

Fig. 5:
The figure shows 'total average precipitation'. It is not clear what the average refers to. Is this the average total precip per month or per day? It is puzzling that adding up all monthly values leads to less than 20 mm per year. A monthly bar plot should show the average cumulative monthly precipitation, the sum over all bars should give the average cumulative annual precipitation.

Fig. 6:
The label 'ZF2' is not explained.

Why do you focus on negative ions only at the T3 site?

Page 11, l. 417:
'The diel cycles of ion and neutral particle concentrations at this site appeared to be very similar in both wet and dry season.'
Is this shown somewhere?

Page 11, l. 425:
'The total particle concentration measured by the MAOS CPC (>10 nm total particle concentration) did not show any diel seasonal cycle.'
Please be precise. Did it not show a seasonal or diel cycle? Or are the diel cycles similar for all seasons?

Page 13, l. 527:
'From the NAIS measurements, a total of 113 days were available for the outside canopy measurements. For the wet season, the data from 28 January until 31 March were used (64 days) and for the dry season the data from 29 August until 13 October was used (46 days).'
Some of these numbers must be wrong.

Page 17, l. 672:
Again, the median rain intensity is according to your figure not highest in the wet season.

Page 17, l. 681:
Do you mean 'while it was' instead of 'where it was'?

In the conclusions you jump between topics (e.g., GR -> air mass origin -> VOCs -> air mass origin).

**Additional technical comments related to the updated manuscript:**

Please carefully revisit your manuscript to check the orthography. There are quite some formal mistakes, which I do not all list here.

Instead of putting 'lat' or 'lon' after geographical coordinations, I suggest to delete the '-' and use 'W' or 'S', respectively.

Why is there no wind velocity but wind direction data for T0t?

In all figure captions with different panels you refer to your panels with 'a', 'b', ... There are no such labels in your figures.

Sometimes you are using abbreviations for months and sometimes not (e.g., page 17, 692).

Caption of table 1:
'The months chosen for the wet season for inside the canopy are Jan-Mar and Dec-Mar for inside the canopy.'
This does not make sense. Also, here you still use inside vs outside canopy terminology.

Table 1:
Make sure you show only significant digits.

Table 1:
The average precipitation values are orders of magnitudes to low for a seasonal average in a rainforest. What do you present here?

Table 3:
The numbers do not make sense. Wet season: 8 NPF days and 57 non-NPF days makes in total 65 days, not 64. Same for rain vs no-rain and for the dry season.\\

Table 4:
Please show only significant digits.

Table 4:
The precipitation rate for NPF days is zero but the average precipitation is 7 mm per day. Please explain how this is possible.

Figure 7:
The legend for the surface plots is incomplete. Generally, the axis labels are too small.

Figure 11:
The SMPS size distribution is plotted on a linear dp axis.

Figure 11:
The NAIS particle concentration time series (4-20 nm) shows gaps but the corresponding surface plot does not - why?

**References**

Kuhn, U., Ganzeveld, L., Thielmann, A., Dindorf, T., Schebeske, G., Welling, M., Andreae, M. O. (2010). Impact of Manaus City on the Amazon Green Ocean atmosphere: Ozone production, precursor sensitivity and aerosol load. Atmospheric Chemistry and Physics, 10(19), 9251–9282. https://doi.org/10.5194/acp-10-9251-2010

Manninen, H. E., Mirme, S., Mirme, A., Petäjä, T., & Kulmala, M. (2016). How to reliably detect molecular clusters and nucleation mode particles with Neutral cluster and Air Ion Spectrometer (NAIS). Atmospheric Measurement Techniques, 9(8), 3577–3605. https://doi.org/10.5194/amt-9-3577-2016

---

## Referee Report (RR2)

I thank the authors for updating the manuscript. Nevertheless, there are still incorrect statements, internal inconsistencies and technical mistakes.
Figures, tables and results were not prepared and discussed carefully.
The updated manuscript does not meet the standards of ACP.

I divide my comments into two major parts. First, I address some author responses.
Afterwards, I address specific comments in the updated manuscript.
I put all author comments by Wimmer et al. in blue italic font.

**Replies to Author comments:**

*All the precipitation days have been excluded in the revised manuscript for the results shown in Tables 2 and 3 and Figures 2 and 3.*

Reply to author comment:
According to table 1, the authors reject minimum 85% of all wet season data. Please justify your results are still representative for an Amazonian wet season.

*We removed pristine from the revised manuscript, as our focus for the dataset presented in our manuscript is not on a pristine environment. Nucleation mode particles have been observed in the Amazon region in the vicinity of Sao Paulo (Backman et al., 2012).*

Reply to author comment:
This statement raises serious concerns. Sao Paulo is some 3500 km SE of Manaus - it is definitely not close to the Amazon region. There is a similar statement in the abstract.

**Additional specific comments related to the updated manuscript:**

I do not list all typos and grammar mistakes - the manuscript needs a careful revision, especially in 2.4 and 3.

The authors mix up their notations for the two different sites. Still you can find 'outside' and 'inside rainforest' or 'rainforest canopy' or 'pasture'.

l. 35: 'The occurrence of NPF on ground level in the Amazon region has been observed previously only in the vicinity of large cities.'
The statement needs justification. If it is related to Backman 2012, it is incorrect.

paragraph 2.4: In this pragraph multiple statements repeat - it needs to be revisited.

l. 316: It is not clear what the authors refer to.

l. 345: 'The intermediate (2-4) positive ion concentrations are about a factor of 2 higher (16 (-) wet, 29(+) wet; 18 (-) wet, 32 (+) dry).' - wet, wet, wet, dry - there is something wrong.

l. 358: The presented numbers don't match those in the respective table.

l. 362: The presented numbers don't match those in the respective table.

l. 415: The Wang 2016 statement is repetitive.

l. 473: 'The results of the back-trajectory calculations are shown in Figure 10. On non-NPF days, the 50 th percentile of air masses originate from about 2.6°S, 56.6°W and 537.4 m.a.s.l., a location on the Amazon river upstream. On NPF days, the back-trajectory calculations show an origin at 1.6°S, 56.5°W and 738.9 mm a. s. l.; further north, which is an area with dense rainforest.'
The quoted air mass origins do not fit to the coordinates in the figure. Also, one should not interpret one single point of a trajectory as the specific origin. Also, 738.9 mm seems wrong.

l. 487: There is no table 4.

l. 532: According to table 1 there were 517 rainy days instead of 646 - which is wrong as well. The yearly sums add up to 643 rainy days.

**Additional technical comments related to figures and tables in the updated manuscript:**

Table 1: Some numbers are incorrectly summed up.
Also, it seems unlikely that there was no rain in March 2012 in the middle of the wet season.

Table 2: The presented numbers are not consistent with the numbers in the text.
Example: the CPC concentration at T3 is according to the text 1000 per cc (2400) for the wet (dry) season. According to the table it is 1000 per cc for both seasons.
Additionally, the median CPC concentration for the wet season is lower than the first quartile (see the table). The in the text indicated 2400 per cc for the dry season are higher than the indicated third quartile. These kind of mistakes occur in all 3 versions of this manuscript which is very concerning.

Table 4 is missing.

There are typos in almost all figure captions and/or labels.

Figure 2: There is a lot of variation during February. This is not discussed.
Additionally, the units are missing. Furthermore, the label states you refer to T0t, the caption states it is the outside forest station.

Figure 3: The units are missing.

Figure 4: The precipitation unit is wrong.

Figure 5: The figure raises a lot of questions. Why is the number of days with and without rain zero for March, why is it even possible that both is zero?
One other example: according to table 1, it was raining on 23 days each in June 2012 and June 2013 but in this figure the number of rain days is between 10 and 15. Figure 5 is totally inconsistent with table 1.
Additionally the unit for precipitation is missing.

Figure 6: In an earlier version, the rain rates at the two stations were comparable. In this version they are different by a few orders of magnitude. It is concerning that the authors do not recognize or discuss this.

Figure 10: Units are missing.

Figure 11: Why is the lower cut off of the SMPS at 20 nm?

---

## Author Response (AR2)

*General comments from Referee 1*
*I thank the authors for addressing my concerns. Nevertheless, there are still open questions, incorrect statements, internal inconsistencies and technical mistakes.*

*The high amount of mistakes in form and content as well as the incorrect citations leave me with serious concerns about the data quality, the analysis and manuscript preparation. The updated manuscript is not improved significantly and does not meet the standards of ACP.*

*A thorough and careful major revision of the data analysis and the entire manuscript is necessary before considering publication.*

General author comments
We thank the referee for the careful revision and concerns about the manuscript. We have carefully revised the manuscript and re-analyzed the dataset we present.

Changes in the current data analysis:
- All the precipitation days have been excluded in the revised manuscript for the results shown in Tables 2 and 3 and Figures 2 and 3.

- The NAIS data have been carefully quality-checked. The NAIS data, especially in the neutral mode, are unreliable due to the multiple charging effect at sizes above 20 nm (e.g. Manninen et al., 2016). In our presented dataset, we observed the most intense noise levels at sizes above 15 nm, so we decided to restrict our data analysis up to 12 nm in the revised manuscript.

- The T3 meteorological data are also derived from a Vaisala system provided by ARM, to make the data more comparable from both sites.

- The definition of the wet and dry season follows now Artaxo et al. (2013) who defined the wet season in the Amazon from January to June and the dry season from July to December. This definition has been applied throughout the whole analysis in the revised manuscript.

- We removed pristine from the revised manuscript, as our focus for the dataset presented in our manuscript is not on a pristine environment. Nucleation mode particles have been observed in the Amazon region in the vicinity of Sao Paulo (Backman et al., 2012). The dataset presented shows aerosol and ion characteristics at measurement sites 40 km and 70 km away from any major anthropogenic pollution sources.

- To address the concerns about the data availability and statistics, we included Table 1, summarizing the NAIS data availability and rain data availability for the whole measurement period.

In order to make it easier to follow the discussion, we address the concerns of the referee that are still open questions from the previous version of the manuscript.

Regarding the comments by the referee concerning the Figures and Tables, we have re-analyzed the whole dataset as described above. Almost all Figures and Tables have changed in the revised manuscript and all the mistakes in the Figures and Figure captions have been thoroughly taken care of. We thank the referee for pointing these mistakes out.

*Referee comment*
*Rephrasing the original sentences does not solve the citation issues. If you rephrase a sentence from a different source, you still have to cite the original source. I am concerned the authors do not take reasonable care to check their citations.*
*Make sure all the identical sentences and text passages mentioned in my first comments are now correctly referenced - even if rephrased.*

Author comment
We take the concern about the referencing and similarities to previous publications very seriously and carefully re-formulated the revised manuscript, including cross-checking all our references.

*Referee comment*
*You mention the term pristine already in the abstract. Your answer 'The term pristine has been removed from the revised manuscript' is incorrect and misleading. Be precise what parts of the manuscript are changed.*
*As referee 2 already pointed out, you have to clearly define 'pristine'.*

Author comment
We removed all the phrases that included pristine in the revised manuscript. This study does not focus on pristine conditions. The comparison of the seasonal characteristics of ions and neutral particles in the Amazonian atmosphere is the emphasis of the presented manuscript.

The effect of the rainforest canopy on the characteristics of ion and neutral particle size distributions in the sub-3 nm sizes have not been studied before. We put our results into this context and highlight the effect of the rainforest canopy on these quantities in our manuscript.

*Referee comment*
*The authors argue that '.. pollution events would appear in the NAIS/SMPS data as elevated aerosol concentrations in the accumulation mode.'.*
*According to Kuhn et al., 2010, the Manaus pollution plume consists to a significant degree of fuel combustion. Kuhn et al., 2010 found CN concentrations up to several 10000 particles per cubic centimeter - the majority of these particles were likely smaller than 40 nm. Hence, pollution events can influence NAIS and SMPS data to highly variable degrees in a broad size range not only in the accumulation mode. The author's argumentation that filter criteria are not necessary is not convincing.*
*Furthermore, in your abstract you insist to present measurements under pristine conditions. You further state, that even the parallel-wind station T0t site is affected by the Manaus pollution plume about once per week. To my understanding, pristine refers to undisturbed, clean or natural conditions.*
*Without a proper filter to exclude pollution sources, your results can not be considered as pristine. Again, please provide a clear definition for 'pristine'.*
*Presenting average values does not help in this case since these average values will be affected by pollution episodes to variable degrees.¨*

Author reply
We agree with the referee in that pollution can influence NAIS and SMPS data in smaller sizes as well. However, we did not filter the results in the revised manuscript for pollution, since our aim in this paper was not to investigate pristine conditions. We present our results at two measurement

sites that are both (60 and 70 km) far away from major pollution sources, which is Manaus in this case. Nevertheless, we are not measuring in pristine conditions, as both sites are to various extents influenced by the Manaus pollution plume.

*Referee comment*
*Even a rough estimate would already help to put your integral DMPS data into context.*
*Are these concentrations underestimated by a factor of 2 or an order of magnitude?*

Author comment
My educated guess is that the DMPS diffusion losses are about 70% transmission at 6 nm and close to 100% at 10 nm for the specific inlet used in this experiment. In the AMAZE-08 experiment where the same aerosol inlet was used, 50% transmission of aerosol particles was achieved at 4 nm.

*Reply to author comment:*
*I do not understand what the authors are trying to say with: 'However, during the daytime the small-scale variability in boundary layer dynamics and in VOC concentrations tends to even out.' Can you please provide references?*

Author comment
The message here is that our analysis does not rely e.g. on eddy covariance techniques that enable identification of flux of VOCs or aerosols in the footprint of the measurements, which depends e.g. on the measurement height and atmospheric lifetime of the compound (e.g. Rinne et al. 2012, ACP). Instead we look into typical concentrations with averaging time of 1 hour. Such averaging masks the small-scale variability in the boundary layer dynamics in the vicinity of the observation site. A similar analysis is done for the VOCs in Wei et al. 2018 (Agric Forest Met).

*Referee comment*
*I have a few questions on this. Does the drift affect positive and negative cluster ions?*
*Furthermore, you relate the drift in cluster ions to ions generated by the corona charger.*
*According to Manninen et al., 2016, all parts of the preconditioning unit are switched off while naturally charged ions are measured. How can then natural ions be affected?*

Author comment
We agree with the referee that this drift in concentrations raises some questions. We looked at the raw NAIS data files to investigate in detail the instrumental performance. The reason for the drift was found to be due to too low sheath filter currents. Not all the ions will be filtered out of the re-circulating sheath air flow, leading to an over-estimation of the ion concentrations.
At T0t site, the sheath filter currents were too high in both polarities after October 7, 2013. The data has been corrected for a factor of 1.8 for the 4 smallest size channels in the NAIS for the time period October 7 to January 7, 2014.
The sheath filter performance was reasonable, after the NAIS was moved to the T3 site, but only in the negative channel. The positive channel is considered unreliable for the whole time period of the measurements at the T3 site and hence no data from the positive ion channel is shown in the revised manuscript. Additionally, the NAIS data from Sept 09, 2014 to Sept 26, 2014 is considered unreliable and has therefore been removed from the analysis.
We included a section describing the phenomenon in the revised manuscript as follows:

We observed an increase in the concentrations of the cluster ions in the NAIS starting from October 7, 2013 to January 21, 2014. By investigating the raw data files, this drift was observed to be due to too low currents in the sheath air filters. The sheath air filters are electrical filters, using corona

needles to neutralize all the remaining ions. The sheath air is re-circulating in the NAIS, hence an inefficient filtering leads to an over-estimation of ion concentrations. The increased ion concentration was due to too low sheath air filter currents in both polarities after October 6, 2013. A correction factor of 1.8 was applied for both polarities in the 4 smallest size channels of the NAIS (0.8-1.25 nm) for the data taken at the T0t site after that.

This increased level in the positive polarity of the natural ions continued when the NAIS was redeployed at the T3 site. The cause was the same (too low a current in the sheath air filters). The negative polarity was performing well at the T3 site. We consider the positive polarity of the natural charged ions in the NAIS at the T3 site unreliable, therefore no data from the positive channel for the T3 site is shown in this study. Additionally, the ion data from 9-26 September 2014 at the T3 site was considered unreliable and also excluded from our analysis.

*Referee comment*
*The authors agree, that from their presented particle number concentrations from NAIS and CPC measurements one could conclude that their findings indicate a dominating nucleation mode in the Amazon aerosol particle number size distribution but at the same time they do not. This is confusing.*
*I would like to outline my concerns based on the results shown in Table 1:*
*1. At T3, the measured particle number concentrations from NAIS (4-20 nm) and CPC (> 10 nm) are on average very similar. If both instruments are comparable this means, almost all or at least a very large amount of the measured particles must be in the size range 10 - 20 nm. Hence, it follows from these measurements, that the aerosol population is dominated by nucleation mode particles. If true, this has to be supported by further size resolved measurements, since it is a stark contrast to the mentioned existing literature (see my first review, e.g., Martin et al., 2010a, Martin et al., 2010b, Zhou et al., 2002).*
*If one cannot draw this conclusion, NAIS and CPC measurements are not consistent. In any case, a detailed paragraph with a thorough discussion on this discrepancy (if it is one) has to be added to this manuscript.*

*2. At T0t this comparison (CPC, NAIS) is not possible. Nevertheless, comparing the particle number concentrations shown in this manuscript (NAIS, 4-20 nm) with total particle number concentrations in the mentioned references, one can again conclude, that the majority of particles is in the nucleation mode size range. The authors argue, that the mentioned references did not focus on new particle formation. But clearly, these authors used instrumentation sensitive to the nucleation mode size range.*
*The authors have to put their findings (at least for the particle concentration data) into context of existing results and have to discuss their high concentration of nucleation mode size particles.*

Author comment
We agree with the referee. We carefully investigated the data quality of the NAIS measurements in our revised analysis. The NAIS is over-estimating neutral particle concentrations, mainly at sizes >20 nm due to the multiple charging effect (Manninen et al., 2016).
To address these issues, we included two new analysis methods in the revised manuscript.

First: we observed most noise in the NAIS in our data at size above 15 nm, for both neutral and ion measurements. Therefore, we decide to restrict our current analysis to the size of 12 nm for both neutral and ion measurements.

Second: For the comparison of the seasonal characteristics at both measurement sites, we excluded all the days with occurring precipitation for the median diel cycles and median numbers shown in Table 2 and 3 and Figures 2 and 3. The concentrations of NAIS neutral 4-12 nm size channel at the T3 site are about a factor of 2.5 (wet) to 4 (dry) lower compared to the CPC concentrations. At T0t the intermediate size range neutral concentrations are even lower ($80\text{-}90 \, \text{cm}^{-3}$) after filtering out the precipitation days and restricting the upper size limit to 12 nm.

We can conclude from these results, that neutral particles are also produced during precipitation (as also shown in Figure 4) and that with our current analysis, the NAIS neutral particle mode is most reliable up to 12 nm.

*Referee comment*
*The rephrased sentence is confusing. How can the rain events be more common in the wet season, when they 'peak' in the transitional season? Also, the reference is unclear. Do you reference Fig. 5 in Martin et al., 2010?*

Author comment
We have also re-analyzed the rain statistics at the T0t site, as shown in Figure 5. The Figure shows the total monthly average precipitation on the right hand-axis and the monthly median number of rain days and no rain days as green and blue bars with the scale on the left-hand axis. The Figure shows a minimum in rain days from July to November and a maximum number of rain days from December to June. This confirms the definition of wet and dry season as presented in Artaxo et al. (2013), as used in the revised manuscript. Also, the total average precipitation shows a minimum between June and November.

*Reply to author comment:*
*The rephrased sentence does not address my main criticism. My main concern is that the 2 orders of magnitude increase for the particle number concentration is not visible at all above the canopy.*
*I agree that the increase in ion concentration will not be detected by the DMPS, but according to you methods section, the DMPS should be able to detect particles larger than 6 nm (taking into account the CPC cutoff and the inlet losses as stated by you). The forest canopy is certainly hindering mixing. Nevertheless, it is likely that a certain amount of these small particles (if generated by droplet splashing) is already produced at the top of the canopy. At least a fraction of these small particles does not have to go through the canopy and should therefore appear in the DMPS measurements.*
*I put up the comparison with the ions produced by biomass burning for another reason. You argue that those ions are able to pass the canopy, but the others are not - why?*

Author comment
We agree with the referee that at least a fraction of those small particles should be produced also above the canopy. Based on our dataset, we cannot make any firm conclusion on the source of the neutral particles above as observed by the DMPS. In the revised manuscript in Figure 2 all the precipitation days are excluded. We can exclude the precipitation as a source of cluster and intermediate ion concentrations at the T0t site.

References

Artaxo, P., Rizzo, L. V., Brito, J. F., Barbosa, H. M. J., Arana, A., Sena, E. T., Cirino, G. G., Bastos, W., Martin, S. T., and Andreae, M. O.: Atmospheric aerosols in Amazonia and land use change: from natural biogenic to biomass burning conditions, Faraday Discuss., 165, 203–235, 2013.

Manninen, H. E., Mirme, S., Mirme, A., Petäjä, T., & Kulmala, M. (2016). How to reliably detect molecular clusters and nucleation mode particles with Neutral cluster and Air Ion Spectrometer (NAIS). Atmospheric Measurement Techniques, 9(8), 3577–3605. https://doi.org/10.5194/amt-9-3577-2016.

Rinne, J., Markkanen, T., Ruuskanen, T. M., Petäjä, T., Keronen, P., Tang, M. J., Crowley, J. N., Rannik, Ü., and Vesala, T.: Effect of chemical degradation on fluxes of reactive compounds – a study with a stochastic Lagrangian transport model, Atmos. Chem. Phys., 12, 4843-4854, https://doi.org/10.5194/acp-12-4843-2012, 2012.

Dandan Wei, Jose D. Fuentes, Tobias Gerken, Marcelo Chamecki, Amy M. Trowbridge, Paul C. Stoy, Gabriel G. Katul, Gilberto Fisch, Otávio Acevedo, Antonio Manzi, Celso von Randow, Rosa Maria Nascimento dos Santos, Environmental and biological controls on seasonal patterns of isoprene above a rain forest in central Amazonia, Agricultural and Forest Meteorology, Volumes 256–257, 2018, Pages 391-406, ISSN 0168-1923, https://doi.org/10.1016/j.agrformet.2018.03.024.

**General author comments**

We thank the referee for the careful revision and concerns about the manuscript. We have carefully revised the manuscript and re-analyzed the dataset we present.

Changes in the current data analysis:

- All the precipitation days have been excluded in the revised manuscript for the results shown in Tables 2 and 3 and Figures 2 and 3.

- The NAIS data have been carefully quality-checked. The NAIS data, especially in the neutral mode, is unreliable due to the multiple charging effect at sizes above 20 nm (e.g. Manninen et al., 2016). In our presented dataset, we observed the most intense noise levels at sizes above 15 nm, so we decided to restrict our data analysis to up 12 nm in the revised manuscript.

- The T3 meteorological data are also derived from a Vaisala system provided by ARM, to make the data more comparable from both sites.

- The definition of the wet and dry season follows now Artaxo et al. (2013) who defined the wet season in the Amazon from January to June and the dry season from July to December. This definition was applied throughout the whole analysis in the revised manuscript.

- We removed pristine from the revised manuscript, as our focus for the dataset presented in our manuscript is not on a pristine environment. Nucleation mode particles have been observed in the Amazon region in the vicinity of Sao Paulo (Backman et al., 2012). The

dataset presented shows aerosol and ion characteristics at measurement sites 40 km and 70 km away from any major anthropogenic pollution sources.

- To address the concerns about the data availability and statistics, we included Table 1, summarizing the NAIS data availability and rain data availability for the whole measurement period.

**Referee comment**

The authors defined pristine by saying "The occurrence of NPF on ground level in the Amazon region has not been observed previously in pristine conditions. In this work, pristine refers to CCN concentrations of a few hundred cm-3." That definition is a little careless. It is quite possible to measure CCN at 0.15% supersaturation and find number concentrations of a 'few hundred cm-3'. At that supersaturation, the particles would be about 100 nm diameter or larger, which for a number concentration of a few hundred/cc is unlikely to be a pristine situation. Also, the authors never define CCN in the manuscript. If you truly mean pristine, then you should rephrase the above sentences as follows: "The occurrence of NPF on ground level in the Amazon region has not been observed previously in pristine conditions, in which the aerosol has not been influenced by anthropogenic pollution." If that is inappropriate, then find a word other than pristine to use.

Author reply

We completely removed the word pristine from the revised manuscript. This particular study does not focus on pristine conditions. The focus of the presented manuscript is to compare the seasonal characteristics of ions and neutral particles in the Amazonian atmosphere. Pristine was mentioned in the previous manuscript as we wanted to point out that nucleation mode particles have only been observed in the Amazon region in the vicinity of Sao Paulo (Backman et al., 2012). We present our results at two measurement sites that are both (40 and 70 km) far away from major pollution sources, which is Manaus in this case. Nevertheless, we are not measuring in pristine conditions, as both sites are to various extents influenced by the Manaus pollution plume.

The second major results in our manuscript focus on the ion and neutral aerosol characterization of the two different measurement locations. The effect of the rainforest canopy on the characteristics of ion and neutral particle size distributions in the very small sizes have not been studied previously, so the manuscript focuses on this characterization.

Referee comment
2) There is a problem with the revised sentence on line 66. Perhaps "…event days are a factor of two lower…"

Author reply
We thank the referee for pointing out the mistake in the manuscript. Since we followed the suggestions by referee1, and thoroughly re-analyzed our dataset, also the text in the revised manuscript has changed significantly.

Referee comment
3) In response to another comment, the authors wrote the following modification: 'T0t is influenced by pollution about once per week, where T3 on the other hand is reached once per day/once per every second day, especially in the afternoon (Martin et al., 2010b supplementary material, Thalmann et al, 2017, de Sa et al, 2017).' Instead of 'where T3 on the other hand is reached once…', I assume you meant to say something like "whereas T3 is impacted about once…"

Author reply
We thank the referee for pointing out this mistake. As we re-wrote the manuscript based on the suggestions of referee 1, also the text in the revised manuscript has changed significantly.

Referee comment
4) The authors missed my point regarding my previous comment "Lines 374-376 and figure 6 – For the ions in the 0.8-2 nm particles, it looks like they simply turn on at rain intensities above 1." You

responded "We made Figure 6 in order to show the relation between rain intensity and ion concentrations. At rain intensities below 1 mm/h the ion concentration especially in the cluster ion size range only contains the natural in background as they are produced via radon decay or galactic cosmic rays. The background cluster ion band can be observed worldwide, yet the concentrations depend on the location as it depends on the sources and sinks for the ions." My point, which I should have made clearer, was that your statement that "some log-linear relation between the ion concentration and rain intensity could be observed for rain intensities >1 mm h-1 for all the three size bins" is incorrect for Fig 6a. In that case, the log-linear relationship for the T0t ion concentrations is not evident: their variation with rainfall appears to turn on about a rainfall intensity of about 10 mm/hr, and it does not exhibit the clear increase with increasing rainfall intensity as it does in the other five plots. Perhaps that is connected to a higher background concentration of smaller ions, but the exception needs to be mentioned. Also, please correct the legend in Figure 6 that refers to ZF2 rather than T0t.

Author comment
We re-analyzed our dataset based on the suggestions of referee 1. Therefore, most of the Figures were changed in the revised manuscript. Figure 6 now shows the maximum negative ion concentrations during precipitation events for both measurement sites as a function of the rain intensity. We added horizontal lines in Figure 6 to indicate the background ion concentrations at the different sites and different cluster sizes that we studied. We removed the sentence about the log-linear relationship as we do not want to make any parameterization of the ion enhancement due to rain.

Referee comment
5) Concerning my comment 18) Figure 7 and lines 385-395, you "rephrased the paragraph in the revised manuscript, line 509-518: 'The 10-20 nm particle concentration showed first a decrease followed by a slight increase up to ~35 cm-3, peaking later than the 6-10 nm particles. However, it is unlikely that these 10- 20 nm particles originate from the same rain-induced burst as seen inside the canopy, as there is no apparent particle growth from the NAIS measurements. It is unlikely that those particles survive until the top of the canopy, as the tree leaves would filter them out. Wang et al. (2016) reported that nucleation mode particles produced in cloud outflows will be transported down with the rain, such that they can be observed at the ground level as an increase in nucleation and Aitken mode concentrations (Dp <50 nm). The appearance of 6-10 nm particles with its peak concentration, could present a similar scenario of small particles brought down from the free troposphere.'" Why is it that 6-10 nm particles going up will be filtered out by the canopy, but 6-10 nm particles going down will make it to the ground: are the downward particles carried in the wake of the rain drops, leaving less time and for diffusion to the vegetation compared with the upward particles? Please elaborate a little on the mechanisms that differentiate the upward- versus downward-moving particles.

Author comment
Based on our dataset, we cannot make any conclusion on the source of the neutral particles above as observed by the DMPS. The ions inside the rainforest canopy produced by the precipitation are very short-lived. Ion concentrations are only increased during the precipitation events and drop to background levels as soon as the precipitation stops.
Figure 2 shows enhanced ion concentrations in the months October to January in the absence of precipitation (all days with precipitation were excluded from the current analysis in Figures 2 and 3 and Tables 2 and 3). Since those concentrations are increased during the dry season months, when local biomass burning is most frequent in the Amazon region, we thought that the source of those could be due to anthropogenic influence. We agree with the referee that based on our dataset, we cannot make a firm conclusion on the source of those ions.

References
Artaxo, P., Rizzo, L. V., Brito, J. F., Barbosa, H. M. J., Arana, A., Sena, E. T., Cirino, G. G., Bastos, W., Martin, S. T., and Andreae, M. O.: Atmospheric aerosols in Amazonia and land use change: from natural biogenic to biomass burning conditions, Faraday Discuss., 165, 203–235, 2013.

Backman, J., Rizzo, L. V., Hakala, J., Nieminen, T., Manninen, H. E., Morais, F., Aalto, P. P., Siivola, E., Carbone, S., Hillamo, R., Artaxo, P., Virkkula, A., Petäjä, T., and Kulmala, M.: On the diurnal

cycle of urban aerosols, black carbon and the occurrence of new particle formation events in springtime São Paulo, Brazil, Atmos. Chem. Phys., 12, 11733-11751, 2012, doi:10.5194/acp-12-11733-2012.

Manninen, H. E., Mirme, S., Mirme, A., Petäjä, T., & Kulmala, M. (2016). How to reliably detect molecular clusters and nucleation mode particles with Neutral cluster and Air Ion Spectrometer (NAIS). Atmospheric Measurement Techniques, 9(8), 3577–3605. https://doi.org/10.5194/amt-9-3577-2016

---

## Author Response (AR3)

We thank the referee for the critical comments.

*Referee comment:*
*Replies to Author comments:*
*All the precipitation days have been excluded in the revised manuscript for the results shown in Tables 2 and 3 and Figures 2 and 3.*
*Reply to author comment:*
*According to table 1, the authors reject minimum 85% of all wet season data. Please justify your results are still representative for an Amazonian wet season.*

**Reply to referee comment:**
We agree with the referee that removing the days with precipitation removes a lot of the available data. The reason why we decided to exclude those days in the previous analysis, was to improve the comparisons of the wet and the dry season. After some internal discussion, we decided to include the precipitation days in the analysis shown in Table 2 and 3 and Figures 2 and 3.

*Referee comment:*
*We removed pristine from the revised manuscript, as our focus for the dataset presented in our manuscript is not on a pristine environment. Nucleation mode particles have been observed in the Amazon region in the vicinity of Sao Paulo (Backman et al., 2012).*
*Reply to author comment:*
*This statement raises serious concerns. Sao Paulo is some 3500 km SE of Manaus*

**Reply to referee comment:**
We are very sorry for this mistake, we changed the statement in the updated manuscript as follows:
*l.35 To our knowledge this is the first direct observation of NPF events in the Amazon region. Previous observations in Brazil showed the occurrence of nucleation mode particles.*

*Referee comment*
*I do not list all typos and grammar mistakes - the manuscript needs a careful revision, especially in 2.4 and 3.*

**Reply to referee comment:**
The main text of the previous version of the manuscript was corrected by a native speaker. Nevertheless, we carefully revised the updated manuscript and corrected the language.

*Referee comment:*
*The authors mix up their notations for the two different sites. Still you can find 'outside' and 'inside rainforest' or 'rainforest canopy' or 'pasture'.*

**Reply to referee comment:**
We thank the referee for pointing out these inconsistencies. The notations for the measurement sites are now consistent as: open pasture/pasture site and inside rainforest site.

*Referee comment:*
*l. 35: 'The occurrence of NPF on ground level in the Amazon region has been observed previously only in the vicinity of large cities.'*
*The statement needs justification. If it is related to Backman 2012, it is incorrect.*

**Reply to referee comment:**

In Backman et al., 2012 the measurement site was located inside Sao Paulo (see Figure 1 in Backman et al.). Nevertheless, we changed the statement in the updated manuscript, to be more precise, as follows:

*l.35: To our knowledge this is the first direct observation of NPF events in the Amazon region. Previous observations in Brazil showed the occurrence of nucleation mode particles.*

*Referee comment:*
*paragraph 2.4: In this pragraph multiple statements repeat - it needs to be revisited.*

**Reply to the referee comment:**

This paragraph is updated in the revised manuscript. We think that the information provided in paragraph 2.4 is essential for the results presented in the manuscript, therefore we keep most of the information. Nevertheless, we changed the wording slightly, to make the paragraph more concise, as follows:

*l. 300: All the available data from the NAIS were cleaned for a potential instrumental noise. The cleaning process was done visually using the particle and ion size distributions as surface plots. Based on this initial screening, the decision was made whether one or more of the electrometers were reliable or not. The non-reliable data were removed based on the guidelines introduced by Manninen et al. (2010). The NAIS data turned out to be unreliable during the measurements presented here mostly in the size range above 15 nm. Therefore, we decided to show data for the sizes up to 12 nm only in our analysis.*

*We observed an increase in the concentrations of the cluster ions in the NAIS starting from October 7, 2013 to January 21, 2014. By investigating the raw data files, this drift was observed to be due to too low currents in the sheath air filters. The sheath air filters are electrical filters, using corona needles to neutralize all the remaining ions, which leads to an over-estimation of the ion concentrations. A correction factor of 1.8 was applied to account for this problem in the 4 smallest size channels of the NAIS (0.8-1.25 nm) for the data taken at the T0t site after the drift was observed.*

*This increased level in the positive polarity of the natural ions continued when the NAIS was re-deployed at the T3 site. The cause was the same (too low a current in the sheath air filters). We consider the positive polarity of the natural charged ions in the NAIS at the T3 site unreliable, therefore the data, regarding the absolute concentrations, using the positive channel for the T3 site is not shown in this study. Additionally, the ion data from September 9-26, 2014 at the T3 site was considered unreliable and also excluded from our analysis.*

*Rain-induced ion events were selected as days, when an ion burst coincided with the onset of precipitation. The median and the maximum (99th percentile) ion concentrations were calculated during periods when the rain intensity was >0 mm h$^{-1}$. In case of more than one rain-event per day, two separate rain-events were classified as such, if the start of the second one occurred more than an hour after the end of the first one. Any fluctuations in the rain intensity for a time period shorter than 1 hour were considered to be part of a single rain-event. At T0t, we classified 962 rain-events and at T3, 221 rain-events.*

*The new particle formation event analysis from the ion spectrometer data, including the event classification and formation and growth rate calculations, followed the already well-defined guidelines (Kulmala et al., 2012). In the data analysis, the first step was to classify all available days into NPF event and non-event days according to methods introduced earlier by Hirsikko et al. (2007) and Manninen et al. (2010). The days which do not fulfill the criteria of an event or non-event day, are categorized as undefined days, however, there were no days classified as undefined in this study. The classification was performed manually through a visual inspection of daily contour plots of particle number size distributions. The second step in the analysis was to define the characteristics related to each NPF event, such as the particle growth rate (GR) and formation rate (J). The GRs were calculated for two different size bins (2-3 nm and 3-7 nm in particle diameter) using both ion and neutral particle data from the NAIS. The particle growth rate was determined by finding the times at which the maximum concentrations of ions/particles in each of these size ranges occurred. A fit between the points was then applied to determine the growth rates. The particle formation rate was determined for the lower end of each size bin (2 and 3 nm) by considering the growth rates, the condensation sink, and the coagulation sink.*

*Referee comment:*
*l. 345: 'The intermediate (2-4) positive ion concentrations are about a factor of 2 higher (16 (-) wet, 29(+) wet; 18 (-) wet, 32 (+) dry).' - wet, wet, wet, dry - there is something wrong.*

**Reply to the referee comment:**
This mistake happened in the process of correcting the language in the previous manuscript. We are sorry for that mistake. It has been corrected in the revised manuscript as follows:
*l. 334: The ion concentrations in the intermediate size range (2-4 nm) are a factor of 2 higher (17 (-) wet, 34(+) wet; 17 (-) dry, 34 (+) dry).*

*Referee comment:*
*l. 358: The presented numbers don't match those in the respective table.*
*l. 362: The presented numbers don't match those in the respective table.*

**Reply to the referee:**
We thank the referee for pointing out theses mistakes, we carefully compared the numbers in the text to those in all the Tables and Figures in the updated manuscript and now there should not be any such mistakes anymore.

*Referee comment:*
*l. 415: The Wang 2016 statement is repetitive.*

**Reply to the referee:**
We agree that the Wang statement is repetitive. We changed the paragraph in the updated manuscript as follows:
*l. 416: The appearance of 6-10 nm size particles and their peak concentration could present a similar scenario as observed in Wang et al (2016) of small particles brought down from the free troposphere. Wang et al. (2016) reported the production of small aerosol particles as a result of new particle formation at cloud outflow region, with further transport within the boundary layer via strong convection during precipitation events in the Amazon. Wang et al. (2016) noted that the <20 nm*

*particle concentrations decreased very rapidly. We suggest the process that we observe to be a local one, as the production of ions was observed to only last for the duration of the precipitation.*

*Referee comment:*
*l. 473: 'The results of the back-trajectory calculations are shown in Figure 10. On non-NPF days, the 50 th percentile of air masses originate from about 2.6°S, 56.6°W and 537.4 m.a.s.l., a location on the Amazon river upstream. On NPF days, the back-trajectory calculations show an origin at 1.6°S, 56.5°W and 738.9 mm a. s. l.; further north, which is an area with dense rainforest.'*
*The quoted air mass origins do not fit to the coordinates in the figure. Also, one should not interpret one single point of a trajectory as the specific origin. Also, 738.9 mm seems wrong.*

**Reply to the referee:**
We agree with the referee that the numbers here are wrong. We corrected the numbers in the updated manuscript

*l. 468 On non-NPF days, the 50th percentile of air masses originate from about 2. 9°S, 58.6°W and 545 m.a.s.l., a location upstream of the Amazon river. On NPF days, the back-trajectory calculations show an origin at 2.5°S, 58.5°W and 602.5 m a. s. l.; further north, which is an area with dense rainforest. The results of the back-trajectory calculations are shown in Figure 10.*

We also agree with the referee that one should not interpret one single point of a trajectory as the specific origin. The back-trajectories were calculated as ensembles and the median values of those were used for presenting the results in Figure 10. We included this in the updated manuscript as follows:

*l. 464: The back trajectories were calculated as ensembles for 24 hours to arrive at 13:00 UTC (09:00 local time) on NPF days at 500m a. s. l.*

*Referee comment*
l. 487: There is no table 4.

**Reply to the referee:**
We are sorry for this mistake, the reference to the table is corrected in the updated manuscript:
*l. 480: Table 3 shows a comparison of the median particle and ion concentrations (25th – 75th percentiles in brackets),*

*Referee comment:*
l. 532: According to table 1 there were 517 rainy days instead of 646 - which is wrong as well. The yearly sums add up to 643 rainy days.

**Reply to the referee:**
The numbers are corrected in the table and text in the updated manuscript. There were 643 total rainy days. The year 2014 in Table 1 is divided into T0t and T3 site.
Table 1 is revised in the updated manuscript, as follows:

| | | # of days with rain data | # of days with rain event | NAIS particle data | NAIS ion data | NPF |
|---|---|---|---|---|---|---|
| 2011 | August | 5 | 2 | 0 | 0 | 0 |
| | September | 6 | 1 | 4 | 4 | 0 |
| | October | 28 | 14 | 31 | 31 | 0 |
| | November | 30 | 18 | 30 | 30 | 0 |
| | December | 31 | 23 | 16 | 16 | 0 |
| total 2011 | | 100 | 58 | 81 | 81 | 0 |
| 2012 | January | 31 | 31 | 31 | 31 | 0 |
| | February | 29 | 18 | 29 | 29 | 0 |
| | March | 31 | 0 | 9 | 9 | 0 |
| | April | 30 | 29 | 29 | 29 | 0 |
| | May | 31 | 25 | 16 | 16 | 0 |
| | June | 30 | 23 | 4 | 4 | 0 |
| | July | 31 | 24 | 0 | 0 | 0 |
| | August | 31 | 12 | 0 | 0 | 0 |
| | September | 30 | 4 | 0 | 0 | 0 |
| | October | 31 | 0 | 0 | 0 | 0 |
| | November | 30 | 5 | 0 | 0 | 0 |
| | December | 31 | 24 | 16 | 16 | 0 |
| total 2012 | | 366 | 195 | 134 | 134 | 0 |
| 2013 | January | 31 | 26 | 31 | 31 | 0 |
| | February | 28 | 28 | 28 | 28 | 0 |
| | March | 31 | 24 | 31 | 31 | 0 |
| | April | 30 | 29 | 30 | 30 | 0 |
| | May | 31 | 27 | 31 | 31 | 0 |
| | June | 30 | 23 | 30 | 30 | 0 |
| | July | 31 | 9 | 31 | 31 | 0 |
| | August | 31 | 15 | 26 | 26 | 0 |
| | September | 30 | 13 | 30 | 30 | 0 |
| | October | 30 | 16 | 31 | 31 | 0 |
| | November | 30 | 24 | 30 | 30 | 0 |
| | December | 31 | 17 | 31 | 31 | 0 |
| total 2013 | | 364 | 251 | 360 | 360 | 0 |
| 2014 | January T0t (rain data only from T0t) | 20 | 13 | 25 | 25 | 0 |
| | January T3 | 0 | 0 | 5 | 5 | 2 |
| | February | 28 | 23 | 28 | 28 | 2 |
| | March | 31 | 28 | 31 | 31 | 4 |
| | April | 30 | 27 | 23 | 23 | 0 |
| | May | 0 | 0 | 0 | 0 | 0 |
| | June | 0 | 0 | 0 | 0 | 0 |
| | July | 0 | 0 | 0 | 0 | 0 |

| | | # of days with rain data | # of days with rain event | NAIS particle data | NAIS ion data | NPF |
|---|---|---|---|---|---|---|
| | August | 31 | 13 | 6 | 6 | 0 |
| | September | 30 | 16 | 30 | 12 | 0 |
| | October | 31 | 19 | 13 | 13 | 0 |
| total 2014 | | 201 | 139 | 161 | 143 | 8 |
| total | | 1031 | 643 | 736 | 718 | 8 |

*Referee comment:*
Table 1: Some numbers are incorrectly summed up.
Also, it seems unlikely that there was no rain in March 2012 in the middle of the wet season.

**Reply to the referee:**
The numbers are now all corrected in the updated Table 1. Our data shows that there was no precipitation in March in 2012, which we agree seems strange, as it is during the wet season.

*Referee comment:*
*There are typos in almost all figure captions and/or labels.*

**Reply to the referee:**
The Figure captions and labels have been carefully revised in the updated manuscript.

Figure 2: There is a lot of variation during February. This is not discussed.
Additionally, the units are missing. Furthermore, the label states you refer to T0t, the caption states it is the outside forest station.

**Reply to the referee:**
The Figure was updated in the revised manuscript, as we now decided to keep the rainy days in the data analysis. The label and the captions are corrected in the updated manuscript. It seems that the variation in February, is a result of excluding the rainy days. As shown in Table 1, we only have data from February from 2012 and 2013. In February 2012, rain occurred on 18 out of 29 days and in February 2013 on 28 out of 28 days, that leaves not many data points in February, when the rainy days are excluded from the analysis. The referee pointed this out correctly, hence we included the rainy days in the analysis in the updated manuscript again.

Figure 2:

[Figure]

Figure 2: The median annual variations for positive and negative cluster (0.8 – 2nm) and intermediate (2 - 4 nm) ions, from the inside the rainforest site are shown. The boxes show the 25[th]-75[th] percentile and the whiskers are 1.5 x IQR (interquartile range), data points beyond the whiskers are considered outliers.

*Referee comment:*
*Figure 3: The units are missing.*

**Reply to the referee:**
Figure 3 is corrected in the updated manuscript.

[Figure]

Figure 3. The median diel patterns of the intermediate (2-4 nm) and the large (4-12 nm) particles from the NAIS measurements at the T0t measurement site are shown. On the left -hand side are the values for the wet and on the right-hand side the values for the dry season. The boxes represent $25^{th} - 75^{th}$ percentiles and the whiskers are 1.5 x IQR (interquartile range), data points beyond the whiskers are considered outliers.

*Referee comment:*
*Figure 4: The precipitation unit is wrong.*

**Reply to the referee:**
Figure 4 is updated in the revised manuscript

[Figure]

Figure 4. An example for a rain event at the T0t, inside the rainforest measurement site is shown. The upper panel shows the surface Figure of the NAIS negative ion channel. The lower panel shows (i) the concentrations of positive (red line), (ii) negative (blue line) cluster ions (0.8 – 2 nm), (iv) positive (dashed black line), and (v) negative (dot-dashed line) intermediate (2 -4 nm) ions on the left - hand axis. The precipitation rate in mmh$^{-1}$ is shown in green on the right - hand axis.

**Referee comment:**
Figure 5: The figure raises a lot of questions. Why is the number of days with and without rain zero for March, why is it even possible that both is zero?
One other example: according to table 1, it was raining on 23 days each in June 2012 and June 2013 but in this figure the number of rain days is between 10 and 15. Figure 5 is totally inconsistent with table 1.
Additionally the unit for precipitation is missing.

**Reply to the referee:**
We agree with the referee that the way the data was presented in Figure 5 in the previous version of the manuscript was not appropriate. In the previous version of the manuscript, we presented the median values for rain and no rain days in the bar plot. This seems to have resulted in the zero values in March. The updated Figure shows the mean number of rain and no-rain days in the bar plot. The data shown in Figure 5 are exactly the same as in Table 1.

[Figure]

Figure 5. The statistics of the precipitation days at the T0t site are shown. The blue bars show the mean number of days per month with no precipitation and the green bars the mean number of days per month with precipitation rates above zero. The black line shows the average total precipitation per month in mm on the right-hand axis.

*Referee comment:*
*Figure 6: In an earlier version, the rain rates at the two stations were comparable. In this version they are different by a few orders of magnitude. It is concerning that the authors do not recognize or discuss this.*

**Reply to the referee:**
We agree with the referee that this should be discussed in the manuscript. In the previous version of the manuscript, the rain data at the pasture site was taken from an optical rain gauge. The rain data in the current manuscript for the T3 site is using also a Vaisala weather station. We thought that the data of the two measurement sites would be more comparable if for the analysis at both sites, Vaisala weather stations are used for the meteorological data. The Vaisala station and the optical rain gauge differ in the precipitation values, the days and times are the same for both data sets. The data from the optical rain gauge and the Vaisala station at T3 are available at the ARM data browser.

A few sentences have been added in the updated manuscript.

*l. 224: In addition to the ion spectrometer measurements, the measurement hut hosted a Vaisala system (WXT-520) for acquiring meteorological parameters.*

*The auxiliary data from the T3 site, presented in this manuscript includes measurements from an ultrafine CPC, with a 50% activation diameter of 10 nm and an SMPS with a lower cut-off of 20 nm. The meteorological data was retrieved from a Vaisala system (WXT-520). Those datasets are available at the ARM data browser.*

*Referee comment*
Figure 10: Units are missing.

**Reply to the referee:**

Figure 10 updated with units in the revised manuscript

[Figure]

Figure 10: median back trajectories for NPF (blue) and non-event (red) days are shown. The trajectories were calculated 24hours backwards arriving at 09:00 local time at 500 m a.s.l. at the open pasture measurement site.

*Referee comment:*
*Figure 11: Why is the lower cut off of the SMPS at 20 nm?*

**Reply to the referee comment:**
When looking at the size distributions measured by the SMPS and comparing them to the ones measured by the NAIS, the SMPS starts to be unreliable below 20 nm. The NAIS is a more reliable instrument for the smaller sizes than the SMPS. In the SMPS, the aerosol needs to be charged first, which is problematic, due to the low charging probabilities at smaller sizes. Additionally, SMPS inlet systems usually suffer from high diffusion losses.

---

## Author Response (AR4)

**Author response**
We thank the Co-editor for the review. The updated manuscript has been carefully revised by
several people and we used the free version of the 'grammarly' software to correct any mistakes
in the grammar.
Please note that we also changed the order of the Figures in order to make the structure of the
updated manuscript more logical. Previous Figure 8 is now Figure 4.

[revised manuscript text omitted]